# Performance Assessment of TanDEM-X DEM for Mountain Glacier Elevation Change Detection

**Julian Podgórski [1,\*], Christophe Kinnard [2] , Michał Pętlicki [3] and Roberto Urrutia [4]**

[1]  Institute of Geophysics, Polish Academy of Sciences, ul. Księcia Janusza 64, 01-452 Warsaw, Poland
[2]  Département des sciences de l'environnement, Université du Québec à Trois-Rivières, 3351, boul. des Forges, C.P. 500, Trois-Rivières, QC G9A 5H7, Canada; christophe.kinnard@uqtr.ca
[3]  Glaciology Laboratory, Centro de Estudios Científicos (CECs), Av. Prat 514, Valdivia 5110466, Chile; michal@cecs.cl
[4]  Centro de Ciencias Ambientales EULA, Universidad de Concepción, Concepción 4089100, Chile; rurrutia@udec.cl
\*  Correspondence: jpodgo@igf.edu.pl

**Abstract:** TanDEM-X digital elevation model (DEM) is a global DEM released by the German Aerospace Center (DLR) at outstanding resolution of 12 m. However, the procedure for its creation involves the combination of several DEMs from acquisitions spread between 2011 and 2014, which casts doubt on its value for precise glaciological change detection studies. In this work we present TanDEM-X DEM as a high-quality product ready for use in glaciological studies. We compare it to Aerial Laser Scanning (ALS)-based dataset from April 2013 (1 m), used as the ground-truth reference, and Advanced Spaceborne Thermal Emission and Reflection Radiometer (ASTER) V003 DEM and SRTM v3 DEM (both 30 m), serving as representations of past glacier states. We use a method of sub-pixel coregistration of DEMs by Nuth and Kääb (2011) to determine the geometric accuracy of the products. In addition, we propose a slope-aspect heatmap-based workflow to remove the errors resulting from radar shadowing over steep terrain. Elevation difference maps obtained by subtraction of DEMs are analyzed to obtain accuracy assessments and glacier mass balance reconstructions. The vertical accuracy ( $\pm$ standard deviation) of TanDEM-X DEM over non-glacierized area is very good at $0.02 \pm 3.48$ m. Nevertheless, steep areas introduce large errors and their filtering is required for reliable results. The 30 m version of TanDEM-X DEM performs worse than the finer product, but its accuracy, $-0.08 \pm 7.57$ m, is better than that of SRTM and ASTER. The ASTER DEM contains errors, possibly resulting from imperfect DEM creation from stereopairs over uniform ice surface. Universidad Glacier has been losing mass at a rate of $-0.44 \pm 0.08$ m of water equivalent per year between 2000 and 2013. This value is in general agreement with previously reported mass balance estimated with the glaciological method for 2012–2014.

**Keywords:** glacier; TanDEM-X; DEM; Chile; Universidad Glacier; elevation change; mass balance; performance

## 1. Introduction

Monitoring of changing ice masses worldwide is presently an important topic due to their significance as climate change indicators [1] and relevance as a water source to millions of people in the world [2]. The melting of glaciers is also a driver of global sea level rise [3], which threatens densely inhabited coastal areas worldwide [4].

An established tool for glacier change detection is the analysis of Digital Elevation Models (DEM), which are computer representations of surface relief. Geometric changes of ice masses reflect the



condition of a glacier with prolonged negative mass balance manifesting itself as frontal retreat and overall thinning, while advance of the front may be a symptom of mass gain or surging [5]. Multi-temporal analyses, using DEMs from different times, can be used to track these changes and provide valuable estimates of ice mass state. The elevation models themselves can be acquired with different methods. Laser scanning, radar imaging and photogrammetric analysis of optical stereoimagery are common ways of creating DEMs.

Laser scanning (also known as Light Detection and Ranging, or LiDAR) is a method increasingly used in cryospheric research [6]. Aerial Laser Scanning (ALS) was first used to map glacierized area by Hopkinson et al. [7]. Examples of use of ALS in glaciology include mass balance analysis of an Icelandic glacier [8] or a change detection study of Austrian glaciers [9], where the LiDAR-based DEMs were used in concert with elevation models built from aerial photographs. While LiDAR is considered a very accurate method of topographic mapping it is expensive when large areas need to be surveyed. Thus, stereoimagery from satellites, aerial surveys or satellite radar altimetry is often a preferred source of elevation data [10].

Currently, there are several global DEM datasets available to geoscientists. The Shuttle Radar Topographic Mission (SRTM) DEM is a radar-based elevation model created in 2000 during the space shuttle mission [11]. Advanced Spaceborne Thermal Emission and Reflection Radiometer (ASTER) DEM is based on the correlation of stereoimages acquired by the ASTER instrument on board the Terra satellite [12]. Images are available from 2000 until present day, as well as a composite global DEM (ASTER GDEM) covering the entire Earth land surface [13]. The ALOS PRISM DEM is another stereoimage-based global DEM, built with data from the PRISM instrument on board the ALOS satellite, active between 2006 and 2011 [14]. These products have a common spatial resolution of 30 m (with SRTM available in 90 m as well) and are commonly used in glaciological change detection studies [15–18].

TanDEM-X is a satellite mission intended to build a global DEM using radar interferometry and demonstrate novel radar imaging methods [19]. The synthetic aperture radar (SAR) data from this mission have been successfully used for studies of glaciers and their changes [20,21], but the users needed to create DEMs from scratch, using the raw radar acquisitions. TanDEM-X DEM is a global DEM product recently released by the German Aerospace Center (DLR). The dataset is based on X-Band SAR acquisitions done by a pair of polar-orbiting satellites [22]. Its exceptional feature, as compared to the other DEMs, is its high spatial resolution of 0.4 arc-second (∼12 m). Its release makes the result of the TanDEM-X mission available to the scientific community, circumventing the need to process radar imagery into DEMs. It is, however, produced by averaging of several DEMs acquired over the years 2011–2014 [23]. The intermediate DEMs (iDEMs) averaged for the final product represent different states of the glacier dependent on season and varying according to a multi-year trend. This casts doubt on the usefulness of TanDEM-X DEMs for change detection studies in glaciology due to the inherent variability of glacier surface elevation, seasonally and over the 3-year sampling period. The iDEMs, however, are not available to the scientific community, thus only the final product TanDEM-X DEM was used in this study.

The objective of this work is to determine the suitability of the TanDEM-X DEM for glacier change detection. This is achieved by comparing the TanDEM-X DEM to other DEMs acquired from various sources, for a case study on Universidad Glacier, an alpine valley glacier in the central Chilean Andes. Performance assessment of TanDEM-X DEM has already been done globally [24] and over sites in South America: Brazil [25] and the Central Andean Plateau [26]. However, studies dealing specifically with the accuracy of the product over glaciers are still lacking. Following our accuracy assessment, we also present geodetic mass balance estimates for Universidad Glacier at the beginning of the 21st century as reconstructed from DEMs from different times and sources.

## 2. Study Area

Universidad Glacier is a valley glacier in central Chile (34°40′ S, 70°20′ W), located circa 130 km south of the country's capital Santiago (Figure 1A). The glacier surface area, as of April 2013, was 27.43 km² [27], which makes it the largest Chilean glacier north of the Patagonian Icefields. The elevation of the glacier ranges between 2450 m a.s.l. at the tongue and a maximum of 4550 m a.s.l. (Figure 1C). The heavily populated area of central Chile depends on rivers for water supply: glacial meltwater can be an important source of water there during droughts and dry summers [28]. One of them is Tinguiririca River, which is fed by Universidad Glacier via the San Andrés tributary river [27]. It was estimated that Universidad Glacier's meltwater contributed up to 20% of flow in the Tinguiririca River basin in 2010 [29]. This value can be expected to rise with continuous warming of the climate in the area [30] and as the glacier mass balance was increasingly negative between 2012 and 2014 [27]. Thus, the results of our work may have societal and economic relevance, aside of the scientific input.

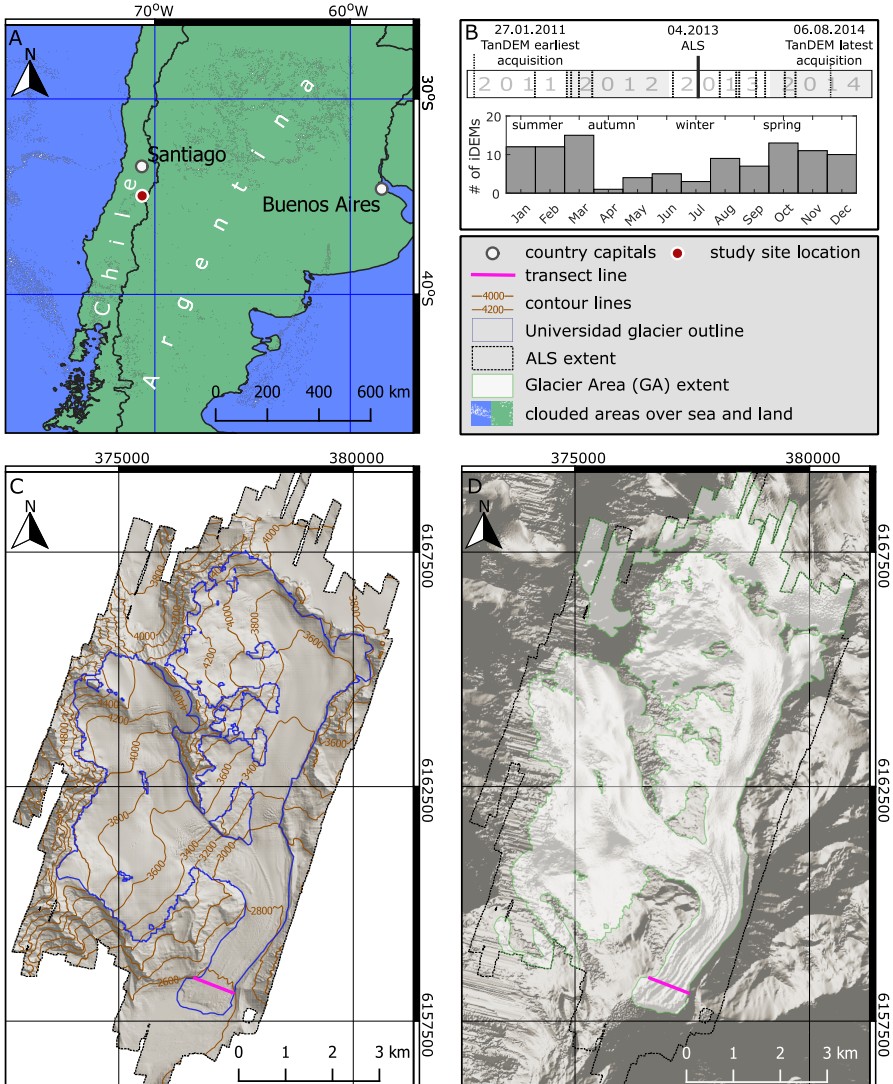

**Figure 1.** Spatial and temporal extent of the study. (**A**)—study site location; (**B**)—temporal distribution of SAR acquisitions used to build TanDEM. The acquisitions dates (dotted lines) were extracted from the DEM metadata. The solid line indicates the acquisition date of the ALS DEM used as reference. (**C**)—Universidad Glacier topography, with hill-shaded ALS DEM used as basemap. The extent of GA is outlined in blue; (**D**)—extent of all glacierized areas within the study area, with hill-shaded TanDEM DEM 12 m used as basemap. Areas falling outside the green outline and within the ALS DEM extent were considered stable area (SA).

## 3. Data and Methods

### 3.1. Data Used

DEMs from four different sources were used for this study: TanDEM-X DEM from the German Aerospace Center (DLR), Aerial Laser Scanning-based DEM acquired in April 2013 (ALS), and 2000 SRTM and 2003 ASTER DEM as representations of the past state of the glacier (Table 1).

**Table 1.** Characteristics of the DEMs used in the study.

| Name | Source | Based on | Original Res. | Res. Used | Time |
|:---:|:---:|:---:|:---:|:---:|:---:|
| ALS DEM (ALS) | This study | Aerial laser scan (ALS) | 1 m | 12 m 30 m | April 2013 |
| TanDEM-X DEM (TanDEM) | DLR | SAR X-band | 12/30 m | 12 m 30 m | Average of 2011–2014 |
| SRTM v3 DEM (SRTM) | NASA [31] | SAR C-Band void-filled | 30 m | 12 m 30 m | 11–22 February 2000 |
| ASTER DEM V003 (ASTER) | NASA JAXA [32] | IR stereopairs | 30 m | 12 m 30 m | 9 April 2003 |

Res.: resolution.

The UTM Zone 19S projected coordinate system was used with the WGS84 ellipsoid as the vertical datum. All DEMs not originally issued in this reference were reprojected in ArcGIS software.

The ALS was used as the reference dataset for corrections and comparisons. The DEM original resolution is 1 m. It was created from ALS point clouds acquired during April 2013 over Universidad Glacier and its immediate surroundings. The scans were done with a Riegl LMS-Q560 aerial laser scanner using full waveform analysis for high-resolution topographic mapping [33]. This type of device has been successfully used before in glaciological contexts, for example in mapping debris-covered glaciers in Iran [34]. The density of the point cloud used was 2 points per square meter and it covered the entire Universidad Glacier and its immediate surroundings (Figure 1) with a vertical inaccuracy lower than 0.1 m. The initial 1 m ALS DEM was aggregated to 12 and 30 m to match the resolution of the other datasets, by taking the mean elevation Z of all 1 m pixels falling within each pixel of the overlying 12 or 30 m grid.

We have received the TanDEM-X DEM from DLR in two resolutions—12 m (TanDEM12) and 30 m (TanDEM30). The aim of the TanDEM-X DEM project is to create an elevation model conformant to the HRTI-3 standard, with 12 m spatial resolution, 2 m of vertical accuracy on flat ($\alpha < 20\%$) terrain and 4 m in steeper areas [19]. The 30 m version is generated from the unweighted mean values of the underlying 12 m pixels [23]. It is intended to fit the DTED-2 standard (30 m spatial resolution, 12/15 m vertical accuracy in the same slope classes) [19]. These restrictive goals are set for point-to-point 90% linear error (LE90) over a $1° \times 1°$ cell, an area larger than our study area. We created a new TanDEM 30 m dataset by bilinearly interpolating the TanDEM12 to the coarser resolution (TanDEM30bili) to compare the impact of the downsampling method on the DEM quality. Additionally, a 90 m TanDEM-X DEM has been recently released by DLR. We have, however, not included it in our study due to the small extent of our region of interest, which would not ensure correct coregistration of the low-resolution DEM to the ALS reference.

DEMs from the Shuttle Radar Topographic Mission [31] and ASTER DEM V003 (ASTER) [32] were chosen as representations of the past state of the glacier because of their popularity for glaciological studies and availability of data taken during the austral summer. The ASTER DEM is based on stereoscopic matching of imagery acquired by the Advanced Spaceborne Thermal Emission and Reflection Radiometer instrument, mounted on the Terra satellite [32]. The chosen ASTER scene was acquired in April 2003 (Table 1) when snow cover was minimal on the glacier. The choice was based on visual interpretation of the multispectral ASTER imagery used for creating the DEMs—care was

taken to ensure that the whole study area was cloudless, as clouds would compromise the quality of the DEM built from stereomatching of the images. Figure 2A shows the study area on a true-color image from ASTER scenes which were used to create the DEM used in the study. No apparent cloud cover is seen over the entire glacier.

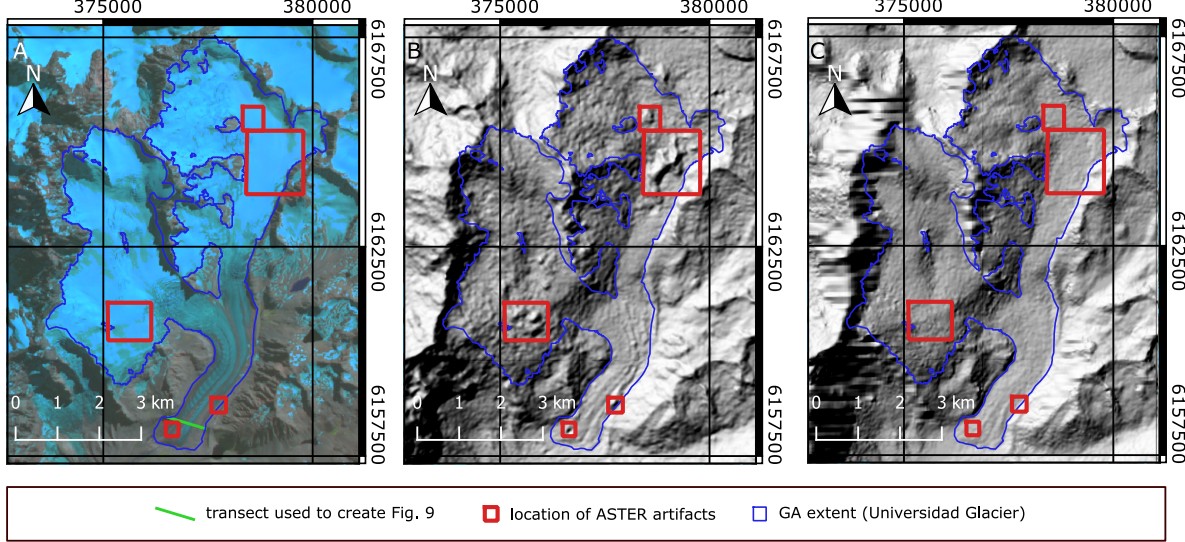

**Figure 2.** Artifacts of ASTER DEM on Universidad Glacier: (**A**): ASTER true-color composition (9 April 2003); (**B**): ASTER DEM hill-shade; (**C**): SRTM DEM hill-shade.

To further exclude the possibility of such error, we compared the ASTER image to the MOD35 cloud mask product based on MODIS acquisitions [35]. The ASTER scene was acquired on 9 April 2003 at 14:52 UTC, while the MOD35 mask used is based on a scene acquired on the same day between 14:50 and 14:55 UTC, ensuring exact temporal overlap. The white pixels on Figure 1A represent areas of "cloudy" and "uncertain clear" classes from the mask. It can be seen that the cloud formations are located far away from the study site, confirming that clouds did not interfere with the creation of the DEM.

The SRTM DEM has been created during a space shuttle mission between 11 and 22 February 2000 and is based on SAR C-Band radar acquisitions. We have used the SRTM v3 version of the model, as it has voids filled with use of ASTER GDEM2 and USGS GMTED2010 elevation models [36]. The voids of the original SRTM were present on areas most troublesome for a radar scanner to image, such as mountain peaks, and were absent on the surface of Universidad Glacier, so the choice of version did not affect the geodetic mass balance results. ASTER and SRTM have an original resolution of 30 m. They were additionally resampled to 12 m with bilinear interpolation to produce higher resolution datasets (ASTER12, SRTM12), which could be directly compared to the TanDEM12 and the finer ALS.

### 3.2. Data Filtering

The study area included the glacierized area and its surroundings, including mountain peaks, valley slopes and the glacier's forefield. A shapefile from the Randolph Glacier Inventory (RGI) [37,38] served to determine the extent of stable area (SA)—the part of the study area not covered by ice and, thus, assumed to not have changed between 2000 and 2013. The timestamp for the Universidad Glacier outline in RGI is given as 2000, the earliest time of our multi-temporal analysis. The outline of Universidad Glacier from the RGI was thus corrected manually to account for debris-covered areas at the tongue and lateral moraines not included in the original contour. Figure 1D presents the whole glacierized area within the study area with a green outline. This area, the extent of Universidad Glacier (GA), was subject to glacier mass balance study. The extent of GA is shown on Figure 1C, while the SA is the area outside of the green-contoured field on Figure 1D. Although it is seen on

Figure 1D that Universidad Glacier is not the only glacierized area within the study area, we have decided to omit the fragments of glaciers in the NE, NW and SW of the study area during mass balance assessment, because these limited fragments are not suitable for geodetic mass balance assessment of the whole glaciers.

Despite their high quality, the TanDEM, ASTER and SRTM contain a certain number of pixels with voids or extreme values (outliers) caused by measurement errors, especially on the rugged surfaces of the mountains surrounding the glacier. These erroneous pixels were filtered out based on the elevation difference ($\Delta Z$) regarding ALS. For the GA, the filtering of data for all applications (glacier mass balance assessment, histogram and semivariogram creation) included two steps: (1) removal of all No-Data pixels, including any voids present in the DEMs and (2) removal of pixels lying outside the 5th–95th percentile intervals. In case of SA, filtering included an additional step in all analyses except for TanDEM systematic error correction (Section 3.4): removal of pixels, where the terrain slope ($\alpha$) > 40°.

### 3.3. Geometric Correction

To correct geometric errors in the TanDEM, SRTM and ASTER products we used the universal method of geometric correction (GC) of DEMs developed by Nuth and Kääb [39]. It uses slope, aspect, and elevation difference maps to compute the 3D misalignment of a DEM relative to another. We used ALS as a reference DEM, as it is the most accurate of the products used. The GC was performed by using pixels from the SA only. It was implemented in MATLAB and all DEMs were subjected to it. Its result is a 3-dimensional translation vector, which, when applied to the analyzed DEM, aligns it horizontally to the reference with sub-pixel accuracy and, additionally, removes the vertical bias. The coordinates of the vectors for each DEM analyzed are given in Table 2.

**Table 2.** Results of corrections of DEMs relative to ALS DEM. Median $\Delta Z$ calculated for DEMs before (Initial) and after (Final) GC, SAHC and filtering to the 5–95 percentile bounds and $\alpha < 40°$. Uncertainties are reported as $\sigma_c$ (Equation (1)). The vertical absolute accuracy (LE90) was computed on the unfiltered dataset after GC and SAHC. All results are based on SA pixels exclusively.

| Dataset | GC Shift Vector (m) | | | Median $\Delta Z$ (m) $\pm \sigma_c$ | | LE90 (m) | | |
|---|---|---|---|---|---|---|---|---|
| | $\Delta$X | $\Delta$Y | $\Delta$Z | Initial | Final | $\alpha < 11.31°$ | $11.31° < \alpha < 40°$ | $\alpha > 40°$ |
| TanDEM12 | 7.55 | 3.33 | −0.32 | −0.11 ± 101.89 | 0.02 ± 3.48 | 1.09 | 16.50 | 215.29 |
| TanDEM30bili | 6.01 | 2.01 | −0.27 | −0.20 ± 100.37 | 0.02 ± 6.36 | 1.35 | 22.73 | 221.98 |
| TanDEM30 | 51.25 | 1.05 | −0.22 | 2.13 ± 106.32 | −0.08 ± 7.57 | 2.03 | 26.77 | 229.65 |
| ASTER | 52.86 | 21.68 | −3.65 | −6.45 ± 61.63 | 0.22 ± 8.90 | 15.59 | 24.54 | 82.82 |
| SRTM | 52.54 | −3.29 | −5.82 | −10.61 ± 53.51 | −0.11 ± 7.31 | 12.56 | 24.88 | 123.04 |
| ASTER 12bili | 16.37 | 23.70 | −4.30 | −4.49 ± 57.65 | 0.41 ± 8.85 | 11.14 | 24.68 | 81.22 |
| SRTM 12bili | 4.23 | −0.39 | −6.16 | −6.30 ± 52.52 | 0.06 ± 7.04 | 6.17 | 24.75 | 120.63 |

### 3.4. TanDEM-X DEM Systematic Error

A glacier is a dynamic system, whose thickness (and thus elevation) is subject to seasonal cycles and multi-year trends. TanDEM12 is a weighted average of several intermediate DEMs built from individual radar acquisitions with weights determined by each iDEM's quality [23]. Figure 1B shows the temporal distribution of intermediate DEMs covering our area of interest. The ALS, taken in April 2013, lies close to the middle of the acquisition period. As the weights used for the creation of TanDEM12 are not publicly available, we assumed them to be equal. In this way the TanDEM12 represents a multi-year average of glacier elevations, which includes observations from all seasons, albeit with a bias towards the warm season, i.e., spring and summer (Figure 1B).

The elevation difference computed between TanDEM and ALS is a sum of three signals: (1) the actual difference between the glacier surface surveyed on April 2013 and the averaged 2011–2014 glacier elevation; (2) the uncertainty resulting from the penetration of the radar signal into snow and firn [40]; and (3) systematic and random errors caused by radar shadowing, a phenomenon common

on steep mountainous terrain. Although the TanDEM-X system can reduce the impact of this error source by using radar acquisitions from both ascending and descending orbits [41], data artifacts are still visible on the DEM (Figure 1D). We had to remove the influence of the latter component to determine the magnitude of the former two signals.

High surface slope ($\alpha$, the angle of inclination of the terrain, 0–90°) and terrain irregularities influence the quality of the TanDEM-X DEM, depending on the viewing angle, as certain areas cannot be sensed due to shadows [22,42]. Therefore, we propose a correction method based on slope-aspect heatmaps (SAHC - Slope-Aspect Heatmap Correction) to correct these problems. Maps of slope ($\alpha$) and aspect ($\Psi$, 0–360°, with 0° being N) were created in ArcGIS with the ALS for the study area and further classified into 1-degree slope bins and 4-degree aspect bins. The relationships between elevation, aspect, and median ALS-TanDEM $\Delta Z$ in each slope-aspect bin was calculated and examined.

A spline surface was fitted to the 3D (aspect vs. slope vs. median $\Delta Z$) heatmap for SA pixels, to model systematic errors related to slope and aspect. The data were filtered to include only bins which represent the slope/aspect distribution of the GA. The filtering ensured that (a) only the pixels relevant for the glacier change detection study were taken into account and (b) that the most erroneous data, which could distort the fitted surface, were excluded. The values of the 2D function fitted on the SA were subtracted from the $Z$ values of the GA at each pixel, according to each pixel's slope and aspect. Figures 3 and 4 panels F-G show the pre- and post-correction median $\Delta Z$ heatmaps of the GA. The heatmap in panel G represents the sum of factors (1) and (2).

After applying GC and SAHC a residual bias remained due to multi-year averaging of iDEMs for the TanDEM final product. We used the median of $\Delta Z$ over SA as estimator of the bias, due to the non-normal distributions of $\Delta Z$ values (Figure 5). This value was subtracted from the corrected DEMs to remove the remaining bias.

### 3.5. Geodetic Mass Balance

The corrected DEMs from the different dates were compared to compute mean glacier thickness change over time. The elevation change was computed as a simple difference between the later and earlier DEM. Numerous outliers prompted us to filter the data. Outliers were removed using the two-step filtering procedure described in Section 3.2.

Mass loss was computed as a product of the mean $\Delta Z$ and the relative density of ice. We assumed ice density for volume-to-mass conversion at $850 \pm 60$ kg m$^{-3}$ after Huss [43], making the relative density factor equal to 0.85. The rate of ice loss is the mass loss divided by the number of years elapsed between the respective DEMs (10 or 13). Uncertainties in ice volume and mass changes were based on the error of the mean of $\Delta Z$ ($\sigma_c$).

In addition, we have analyzed the distribution of $\Delta Z$ with elevation. GA maps were classified into 100 m-wide bands of elevation above sea level based on SRTM12. Within each band the $\Delta Z$ values from each map were averaged.

### 3.6. Uncertainty Measurement

We used the standard error of the mean ($\sigma_p$) as an estimate of uncertainty on calculated mean $\Delta Z$ over the GA and all values derived from it (Table 3). The uncertainty of the mean can vary, depending on whether the measurements within the dataset are assumed to be entirely spatially correlated ($\sigma_p = \sigma_c$, i.e., the spatial standard deviation), entirely spatially uncorrelated ($\sigma_p = \sigma_u$, i.e., the standard deviation of the dataset divided by the square root of the number of measurements) or partially correlated. In the latter case the $\sigma_p$ depends on the autocorrelation range of the individual grid points.

We assumed that the DEMs are partially correlated datasets and used the geostatistical method developed by Rolstad et al. [44] to compute $\sigma_p$. The method takes into account the spatial correlation of elevation measurements across the region of interest. First, an empirical semivariogram is created from a $\Delta Z$ map. A theoretical spherical semivariogram is then fitted to the empirical semivariogram.

An interior-point fitting algorithm was used to minimize the sum of squares of residuals between the fitted model and observed points, weighted by the number of observations included in each point. The procedure was executed in MATLAB. Only SA data were used for $\sigma_p$ calculation, with outliers and high-slope pixels filtered out as described in Section 3.2.

**Table 3.** Results of DEM subtraction and ice loss calculations for GA. $\sigma_c$ and $\sigma_u$ were calculated from $\Delta Z$ histogram (Figures 5–7). $\sigma_p$ was calculated on stable area (SA; Table 2).

| Subtracted Datasets | Years | Resolution (m) | Mean Mean $\Delta Z$ (m) $\pm \sigma_p$ | | Ice Loss (m w.e.) $\pm \sigma_p$ | Ice Loss Rate (m w.e.a$^{-1}$) $\pm \sigma_p$ |
|---|---|---|---|---|---|---|
| ALS-TanDEM12 | 2013–2013 | 12 | −0.04 | ± 0.45 | −0.04 ± 0.38 | - |
| ALS-SRTM12 | 2013–2000 | 12 | −6.77 | ± 0.34 | −5.75 ± 0.28 | −0.44 ± 0.08 |
| ALS-ASTER12 | 2013–2003 | 12 | −15.04 | ± 1.33 | −12.79 ± 1.13 | −1.28 ± 0.36 |
| TanDEM12-SRTM12 | 2013–2000 | 12 | −6.40 | ± 0.54 | −5.44 ± 0.46 | −0.42 ± 0.13 |
| TanDEM12-ASTER12 | 2013–2003 | 12 | −14.51 | ± 1.40 | −12.33 ± 1.19 | −1.23 ± 0.38 |
| ALS-TanDEM30bili | 2013–2013 | 30 | −1.05 | ± 3.40 | −0.89 ± 2.89 | - |
| ALS-TanDEM30 | 2013–2013 | 30 | −0.23 | ± 3.94 | −0.19 ± 3.35 | - |
| ALS-SRTM | 2013–2000 | 30 | −6.99 | ± 0.42 | −5.94 ± 0.36 | −0.46 ± 0.10 |
| TanDEM30-SRTM | 2013–2000 | 30 | −6.49 | ± 3.96 | −5.52 ± 3.37 | −0.42 ± 0.93 |
| TanDEM30bili-SRTM | 2013–2000 | 30 | −5.66 | ± 3.42 | −4.81 ± 2.91 | −0.37 ± 0.81 |
| ALS-ASTER | 2013–2003 | 30 | −15.71 | ± 1.07 | −13.36 ± 0.91 | −1.34 ± 0.29 |
| TanDEM30-ASTER | 2013–2003 | 30 | −15.02 | ± 4.08 | −12.77 ± 3.47 | −1.28 ± 1.10 |
| TanDEM30bili-ASTER | 2013–2003 | 30 | −14.13 | ± 3.65 | −12.01 ± 3.10 | −1.20 ± 0.98 |

To ensure that the sill of the model is correctly predicted, the fitting was done iteratively, starting with a small subset of points ($r < 300$ m) and increasing the threshold for inclusion of points at each iteration by 50 m. In this way the values for the sill-factor ($s$), nugget ($n$) and range ($r$) are obtained. These values were used with the equation from Rolstad et al. [44] to calculate the standard error. The different measures of uncertainty in this paper were thus computed as follows:

$$\sigma_c = \text{std}(\Delta Z) \tag{1}$$

$$\sigma_u = \frac{\sigma_c}{\sqrt{N}} \tag{2}$$

$$\sigma_p^2 = \begin{cases} 0 & L \leq \Delta h \\ n\left(\frac{\Delta h^2}{L^2}\right) + s\left(1 - \frac{L}{h} + \frac{1}{5} \cdot \left(\frac{L}{h}\right)^3\right) & \Delta h < L < r \\ n\frac{\Delta h^2}{L^2} + \frac{1}{5}s\frac{r^2}{L^2} & L > r \end{cases} \tag{3}$$

$$\Delta h = \frac{\Delta x}{\sqrt{\pi}} \tag{4}$$

where $\Delta x$ is the spatial resolution of the dataset, $L$ is the radius of a circular region with area equal to the region area, over which the $\Delta Z$ is averaged—corresponding to the sum of pixels taken into account (after filtering) multiplied by the surface area of a pixel (144 or 900 m$^2$) and $N$ is the sample size (number of pixels). We use SA-based $\sigma_c$ in Table 2 to enable comparisons of precision to other published results. In Table 3, $\sigma_p$ is used as measure of uncertainty for the mean glacier elevation change and all measures derived from it, to account for spatial correlation.

The accuracy goals of TanDEM-X DEM are set by the HRTI-3 for TanDEM12 and DTED-2 for TanDEM30 [19]. Within them the accuracy is reported as linear error with 90% confidence interval (LE90). This means that over the area of interest 90% of absolute $\Delta Z$ measurements need to fall below the set goal [45]. We have calculated LE90 over the SA to compare it with the desired goals. The calculation was done by sorting the unfiltered, absolute $\Delta Z$ values ($|\Delta Z|$) of a DEM and determining the value of the 90th percentile of the $|\Delta Z|$ set. We did this in $\alpha$ intervals of 0–11.31°, 11.31°–40° and >40°. The thresholds were determined by HRTI-3/DTED-2 standards that use $\alpha = 20\%$ as a threshold between two slope bands for which requirements are different [19]). The 20% of inclination corresponds to 11.31 degrees. The second threshold of 40° is dependent on our quality assessment of TanDEM (Figures 3D and 4D).

## 4. Results

### 4.1. Data Correction and Quality Assessment

The misalignment of TanDEM12 relative to the ALS DEM is small (Table 2), corresponding to an error of less than one pixel. The 30 m DEM generated from TanDEM12 keeps the good alignment, but all three DEMs initially supplied at 30 m resolution had to be shifted by an order of magnitude greater than the horizontal and vertical vectors to achieve low median $\Delta Z$.

The global vertical accuracy, defined as median SA $\Delta Z$, of both TanDEM is good, as shown by the low magnitude of the Z component of the shift vector (Table 2). The misalignment of TanDEM30 is comparable to that of SRTM and ASTER. The initial median $\Delta Z$ of TanDEM is lower than that of ASTER and SRTM, but the $\sigma_c$ is higher. This hints at the larger number of outliers in the TanDEM datasets, as compared to the other DEMs. After corrections and filtering TanDEM12 scores best, with both median $\Delta Z$ and $\sigma_c$ lowest among all DEMs. While the error of all DEMs is below 0.5 m, ASTER has a higher median $\Delta Z$ than the remaining datasets, while $\sigma_c$ is consistent between nearly all datasets. Only TanDEM12 stands out, with a standard deviation roughly half that of the other models. TanDEM30bili kept the low bias of TanDEM12, but its $\sigma_c$ increased to a level similar, but still lower, than that of the other 30 m DEMs. Interpolation of SRTM to 12 m did not impact the dataset negatively, with accuracies remaining similar to the original resolution both before and after the GC. In the case of ASTER, the median $\Delta Z$ increased for the finer resolution, but $\sigma_c$ remains at the same level.

Both TanDEM respond well to the universal coregistration method, as seen on Figures 3A–C and 4A–C. TanDEM30 benefited particularly from the GC procedure. The $\Delta Z$ was distributed almost uniformly in the uncorrected SA dataset, while GC brought the bulk of the pixels to a near-zero difference. Nevertheless, the histogram remains wider for the coarser dataset, as reflected in the higher $\sigma_c$ value. Despite the GC, regions of high $\Delta Z$ on TanDEM remained on the steep mountain slopes surrounding the glacier. $\Delta Z$ skyrockets for slopes > 40° (Figures 3D and 4D), an effect visible also on the error heatmap in the slope-aspect ($\alpha - \Psi$) domain over the SA (Figures 3E and 4E). Nonetheless, such high-slope areas are uncommon over the studied glacier (Figures 3H and 4H), so that the extreme $\Delta Z$ had a minimal impact on the results of the geodetic mass balance study.

The heatmap shapes and values differ between TanDEM12 (Figure 3E–G) and TanDEM30 (Figure 4E–G). Also, the spline-surface correction results (Figures 3F,G and 4F,G) show relatively little change in the heatmap shape between the uncorrected and corrected GA pixels. The most prominent difference is the change of values in the right (high-slope) part of the graph from a set of both high and low values to a more uniform, highly negative median $\Delta Z$. The upper (facing NW and N) part of the heatmap remained almost unchanged, while the steep ($\alpha > 30$) sections facing NE saw the largest alterations. The 2D histogram of pixel frequency shows that Universidad Glacier is mainly concentrated in low ($\alpha < 3°$) slope and aspects between 120° and 240°—meaning that the glacier is generally facing S and SSW/SSE (Figure 3H).

Filtering and corrections of the TanDEM, particularly the removal of high-slope pixels, markedly improved its accuracy (Table 2). Both median $\Delta Z$ and $\sigma_c$ of TanDEM30 are higher than those for TanDEM12, pointing to the loss of accuracy when downsampling to the lower resolution, regardless of the interpolation method. TanDEM30bili scores slightly better than TanDEM30 on both median $\Delta Z$ and LE90 measures. The bias of both TanDEM30 versions is higher than that of ASTER and SRTM for the same resolution, possibly indicating problems resulting from multi-year averaging or unfiltered pixels affected by slope-related errors. Despite this slightly higher bias, the TanDEMs have a much lower linear error than the other DEMs. Resampling of ASTER and SRTM to 12 m improved LE90, but introduced bias to the models, visible after the GC.

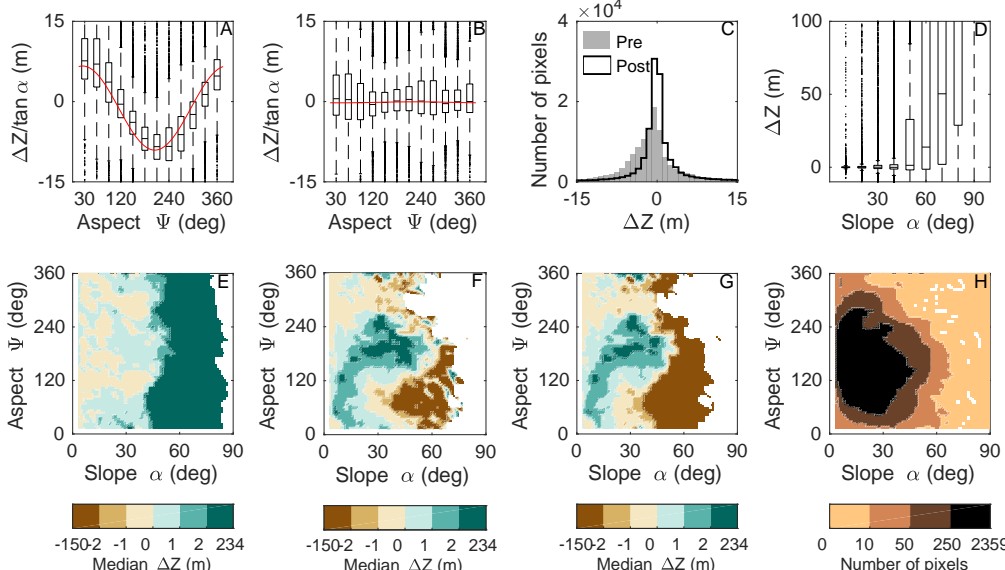

**Figure 3.** Results of data analysis and corrections of TanDEM12 relative to the ALS on SA (**A**–**E**) and applied to GA (**F**–**H**). (**A**,**B**): distribution of $\Delta Z \tan(\alpha)$ vs. aspect $\Psi$ before and after GC respectively; (**C**): $\Delta Z$ histograms before and after GC; (**D**): variation of $\Delta Z$ with slope of the surface. (**E**–**G**): heatmaps of median $\Delta Z$ in slope-aspect ($\alpha - \Psi$) domain binned in $1°$ $\alpha$ and $4°$ $\Psi$ bins. (**E**): SA; (**F**): GA before slope/aspect correction; (**G**): GA after SAHC; The extreme values of the color scale correspond to maximum and minimum median $\Delta Z$ across all three heatmaps. (**H**): 2D histogram of the slope and aspect distribution across the GA.

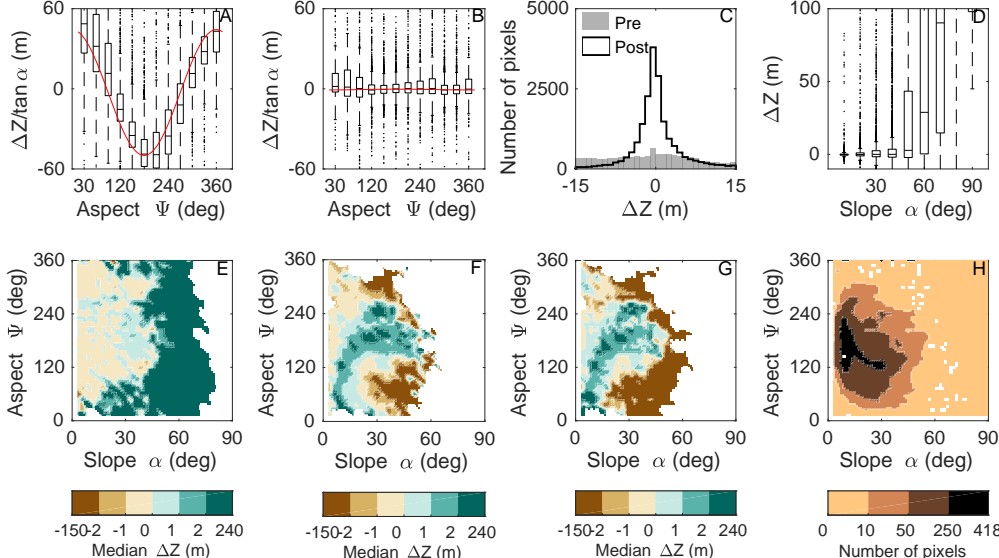

**Figure 4.** Results of data analysis and corrections of TanDEM30 relative to the ALS on SA (**A**–**E**) and applied to GA (**F**–**H**). (**A**,**B**): distribution of $\Delta Z \tan(\alpha)$ vs. aspect $\Psi$ before and after GC respectively; (**C**): $\Delta Z$ histograms before and after GC; (**D**): variation of $\Delta Z$ with slope of the surface. (**E**–**G**): heatmaps of median $\Delta Z$ in slope-aspect ($\alpha - \Psi$) domain binned in $1°$ $\alpha$ and $4°$ $\Psi$ bins. (**E**): SA; (**F**): GA before slope/aspect correction; (**G**): GA after SAHC; The extreme values of the color scale correspond to maximum and minimum median $\Delta Z$ across all three heatmaps. (**H**): 2D histogram of the slope and aspect distribution across the GA.

The ALS-TanDEM $\Delta Z$ over the GA is greater than over the SA for both resolutions (Tables 2 and 3). This might be expected, as the glacier is a dynamic system, contrary to the surrounding mountain

terrain. Thus, multi-temporal averaging of iDEMs is expected to result in an average divergence from the April 2013 surface on the more dynamic part of the scene compared to the stable zone. A greater difference in GA mean $\Delta Z$ is visible when the results of TanDEM30bili and the original TanDEM30 are compared (Table 3). Although the difference of median $\Delta Z$ over SA between the two datasets is only 0.1 m (Table 2), there is almost 1 m of difference between mean $\Delta Z$ of these DEMs on the glacier.

### 4.2. Glacier Geodetic Mass Balance

The spatial distribution of $\Delta Z$ values of TanDEM12 and TanDEM30, relative to ALS, is quite similar (Figure 5). Slightly more negative values appear in the western part of the glacier's northern basin on the TanDEM30 $\Delta Z$ map. The distribution of $\Delta Z$ is different on GA and SA for a given TanDEM product, as evidenced by the different histogram shapes in the GA and SA panels of Figure 5A,B. The trend is mirrored on the maps. A line dividing regions of negative $\Delta Z$ values to the east and positive $\Delta Z$ values to the west can be seen running SW-NE, cutting across the larger, western basin. The bulk of pixels on the GA histograms have positive values, but the tail of the distribution is heavier on the left (negative) side corresponding to red areas on the western margin of the western basin. The SA distribution is more symmetrical, particularly in the case of TanDEM12, reflecting more equal areas of positive and negative $\Delta Z$. On the histogram of ALS-TanDEM30 difference (Figure 5B) the negative tail is present also on the GA graph, suggesting deterioration of accuracy on the $\Delta Z < 0$ areas with interpolation.

The multi-year $\Delta Z$ results are similar when either the ALS or TanDEM is compared with the same old reference DEM (i.e., SRTM, or ASTER). However, the results are markedly different when changing between either SRTM and ASTER as reference DEM. The calculated changes obtained from SRTM (Figure 6) are smaller than those from ASTER (Figure 7) despite the shorter time interval for ASTER (10 years) than for SRTM (13 years). The resolution of the older DEMs seems to impact the results in a limited way, judging by the minimal difference between the 12 vs. 30 m glacier mass balance estimates. Therefore, we have decided to use the 12 m DEMs for final glacier analyses, as they can be paired directly with the more accurate TanDEM12 and 12 m ALS.

The glacier elevation change maps differ slightly in the higher parts of the glacier depending on whether ALS or TanDEM is used as the reference 2013 DEM and compared to older DEMs. On ALS-derived maps the thinning observed in the northern basin is more consistent (Figures 6A and 7A) than on TanDEM-based maps, where patches of $\Delta Z > 0$ are more pronounced (Figures 6B and 7B). The overall thinning pattern is consistent below the accumulation zones, particularly over the icefalls and lower tongue. Ice loss over the glacier's western part is in turn less pronounced on the ALS-based map than on the TanDEM-based one. The mean $\Delta Z$ since either 2000 (SRTM) or 2003 (ASTER) is more negative when TanDEM is used as reference for 2013 than when ALS is used. This is in line with the slightly negative ALS-TanDEM calculated over GA (Table 3 and Figure 5).

As reported for the areal average (specific) mass balances listed in Table 3, the choice between SRTM and ASTER has the most influence on the change detection results. The difference maps based on ASTER (Figure 7) contain large sectors where elevation changes are greater than $-40$ m, whereas the corresponding changes over the same sectors of the SRTM-based difference map appear more subdued and more spatially homogeneous (Figure 6). While the most negative values are found over the glacier tongue on the SRTM-based maps, they are found at much higher altitudes on the ASTER-based maps. Overall, the ASTER maps present more negative $\Delta Z$ values than SRTM, as evidenced by the mean and shape of their corresponding GA $\Delta Z$ histograms (Figure 6 vs. Figure 7). On the other hand, the peak of the SA ASTER histogram is positive, while for SRTM the probability mass is rather concentrated around 0. The divergence between the ASTER and SRTM $\Delta Z$ distributions over SA is however less striking than over the GA.

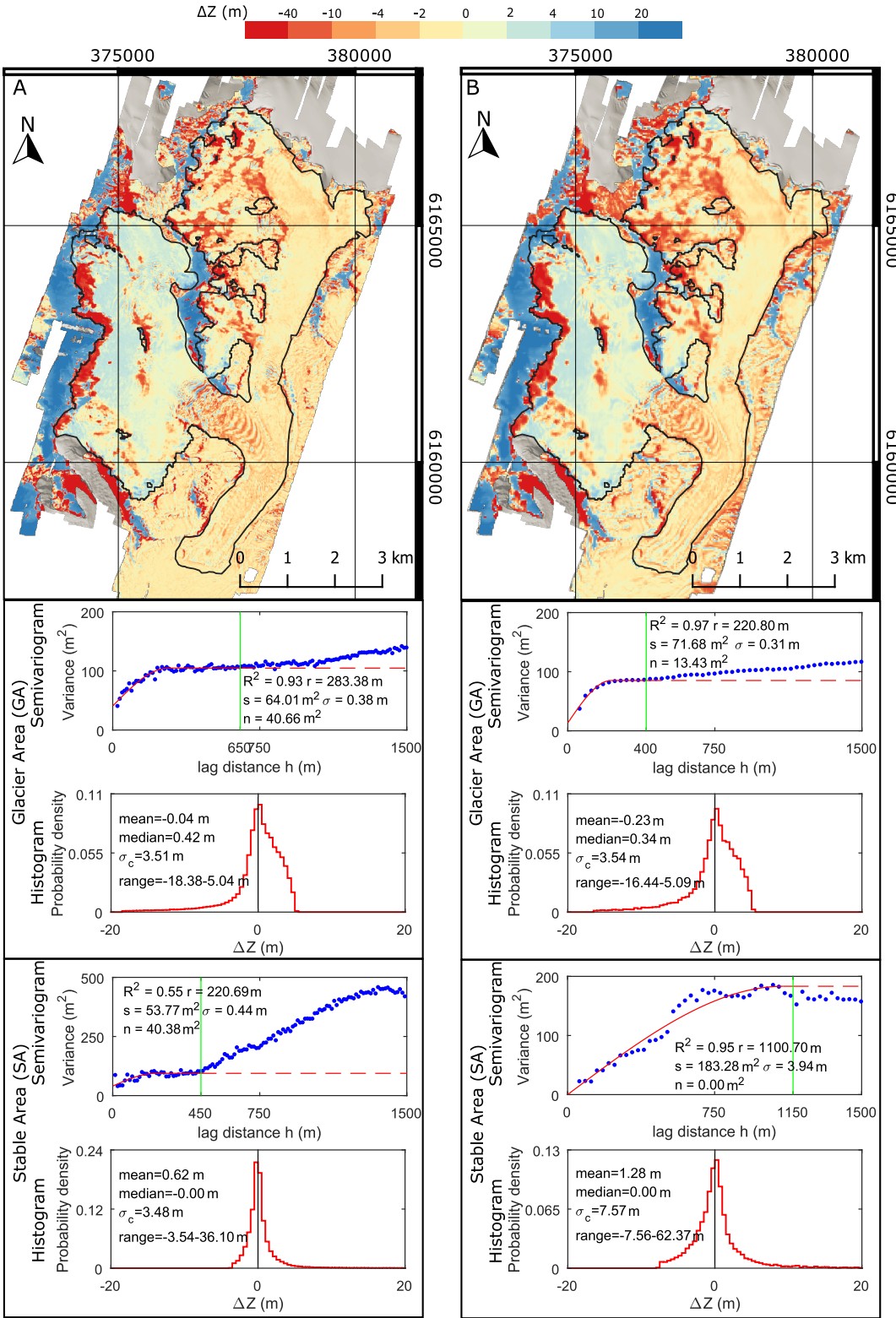

**Figure 5.** Results of DEM subtraction with semivariograms and histograms of differences. (**A**): ALS-TanDEM 12 m; (**B**): ALS-TanDEM 30 m. Blue dots represent empirical semivariograms, while red lines—theoretical ones. Green vertical lines indicate distance thresholds of the best fit of the model to the data. Black line on the map delimits the GA.

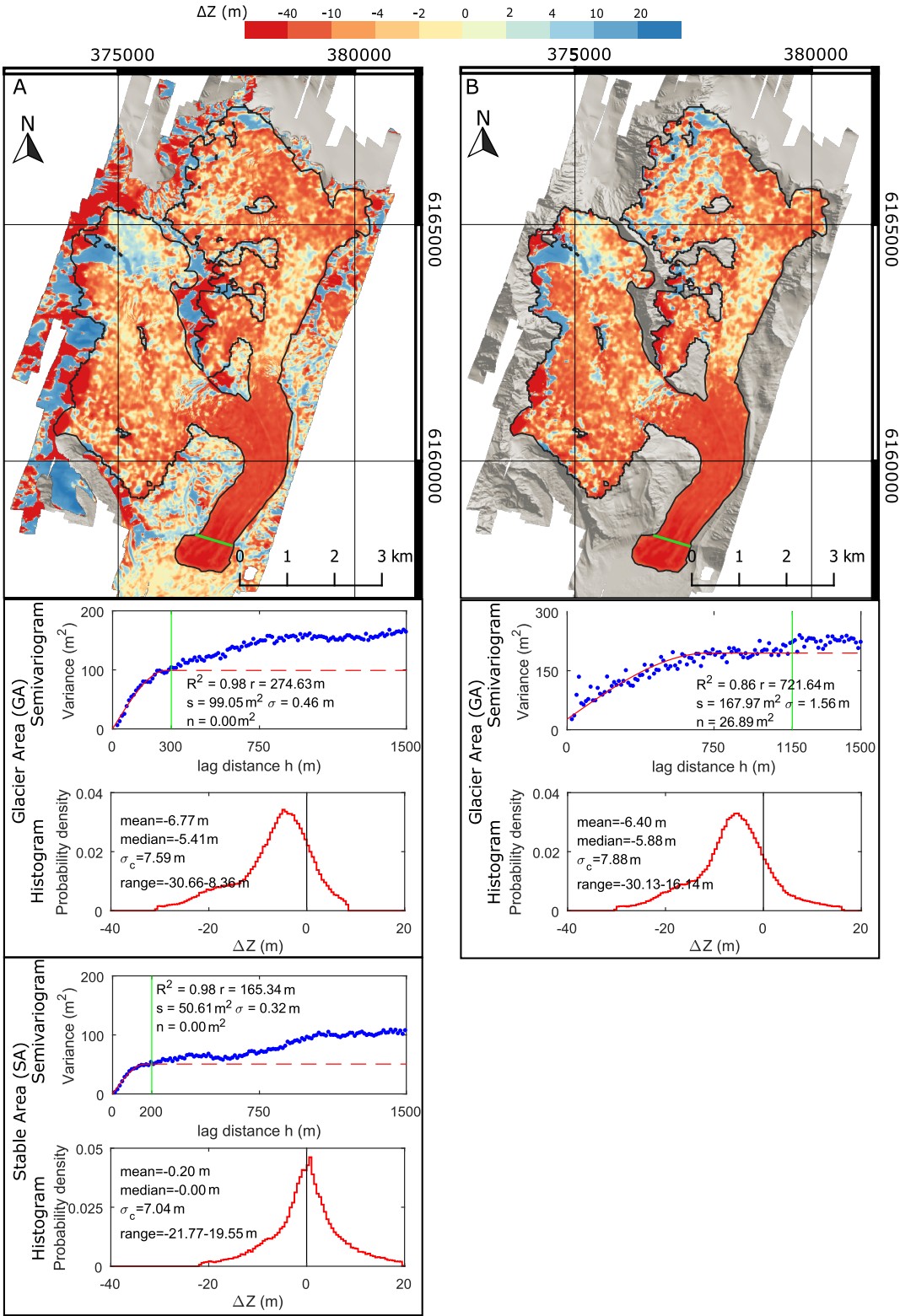

**Figure 6.** Results of DEM subtraction with semivariograms and histograms of differences. (**A**): ALS-SRTM; (**B**): TanDEM-SRTM. Resolution of both maps is 12 m. Blue dots represent empirical semivariograms, while red lines—synthetic ones. Green vertical line indicates distance threshold of the best fit of the model to the data. Black line on the map delimits the GA, green line corresponds to transect line used for creation of Figure 8.

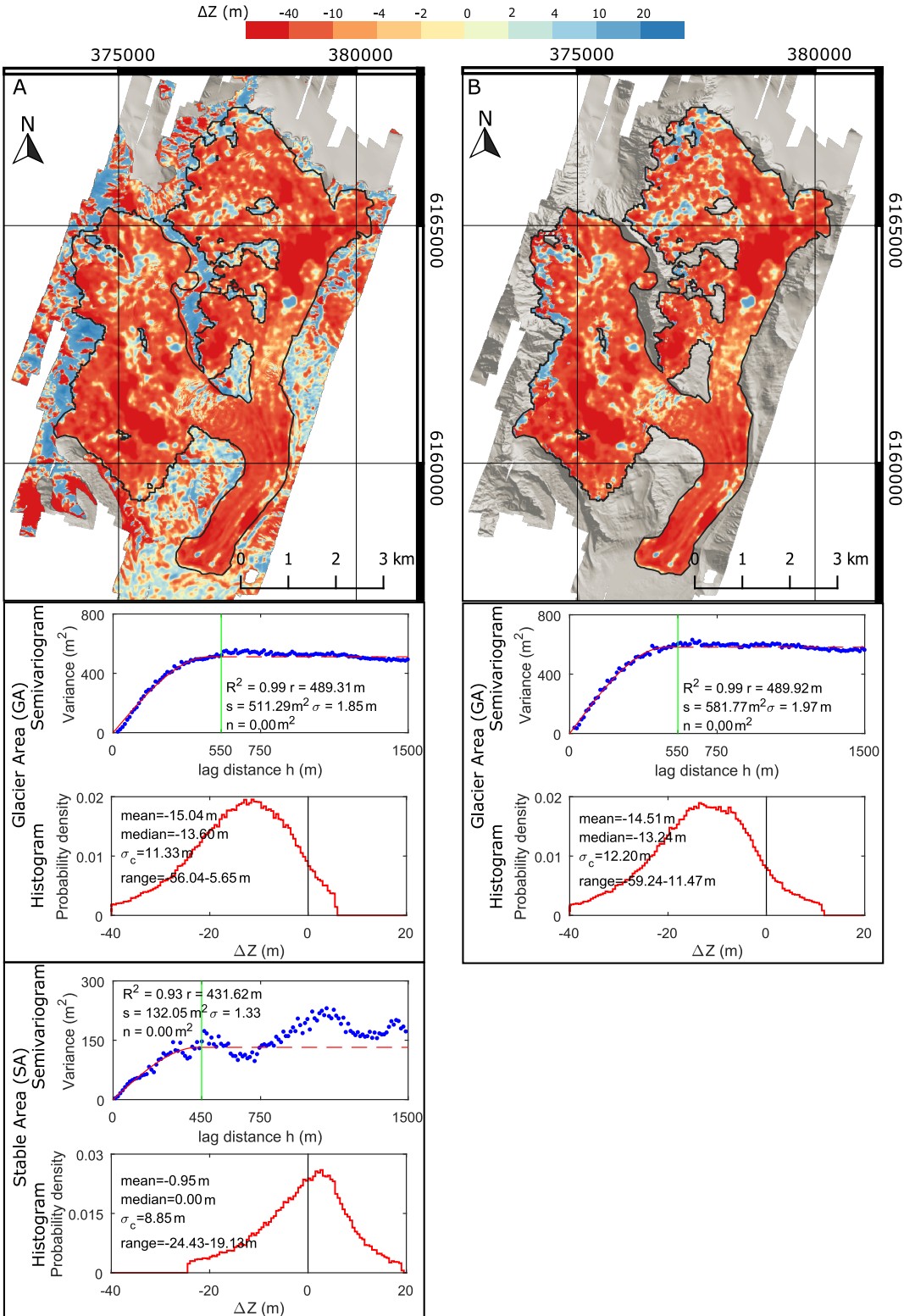

**Figure 7.** Results of DEM subtraction with semivariograms and histograms of differences. (**A**): ALS-ASTER; (**B**): TanDEM-ASTER. Resolution of both maps is 12 m. Blue dots represent empirical semivariograms, while red lines—synthetic ones. Green vertical line indicates distance threshold of the best fit of the model to the data. Black line on the map delimits the GA.

The upward drift visible on the empirical variogram of ΔZ over GA (Figure 6A) is caused by the spatial trend of decreasing ΔZ with increasing elevation, a phenomenon related to the large scale thinning rate pattern over the glacier, i.e., higher at the tongue and lower in the accumulation basins. This trend is not present on the ASTER-derived maps (Figure 7A,B). Over stable areas both cases (ALS-SRTM, ALS-ASTER) show an increase in variance with separation distance, reflecting the contrasted distribution of ΔZ: strongly positive values are found in the westernmost part of the SA and between the two accumulation basins of the glacier, where the relief is steep, while the flat glacier forefield presents values between −4 and 0 m. The same effect, even more pronounced, is seen on ALS-TanDEM maps (Figure 5). On the other hand, the drift of the semivariogram is lower over the GA in these cases, reflecting the more uniform spatial distribution of ΔZ values between the two 2013 DEMs. The SA theoretical semivariograms flatten at ranges below 500 m, but two plateaus can be seen in the empirical data. We assume the long-scale plateau, appearing at lag distances over 1000 m to represent the difference between low-ΔZ forefield and high-ΔZ peaks of the glacier and thus not important for estimating the spatial correlation for standard error assessment. Summary of mean ΔZ change with altitude shows that the glacier has thinned in all altitude bands except one (Figure 9). The largest elevation change is observed at the lowest altitudes (glacier terminus). A steep decrease in absolute ΔZ is observed between 3100 and 4000 m a.s.l., corresponding to the tongue and icefalls. The highest part of the glacier has the lowest absolute elevation changes. The direction of change differs depending on whether TanDEM or ALS is used as the later DEM. TanDEM-SRTM shows thickening of the glacier, while ALS-SRTM ΔZ is positive in only a single elevation band.

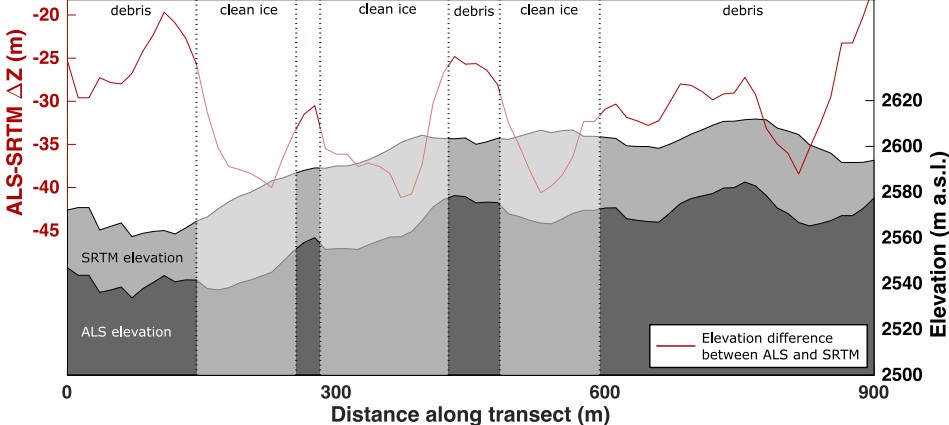

**Figure 8.** Cross-sectional profile of the Universidad Glacier surface elevation in 2000 and 2013 (see profile location in Figure 6).

Thinning rates over the debris-rich surfaces on the glacier's lower part are apparently lower than over clean ice, as seen when comparing maps on Figures 2A and 6. Z and ΔZ values were extracted from ALS, SRTM and ALS-SRTM maps along a transect running across the lower tongue of Universidad Glacier (Figure 8). Troughs on the glacier surface are visible in the ALS elevation plot, which are not present on the SRTM elevation plot. The locations of these troughs overlap with areas of greater thinning and correspond to clean-ice zones on multispectral image (Figure 2A). Between them plots of ALS and SRTM elevation follow paths similar to each other, albeit shifted in elevation due to the general thinning.

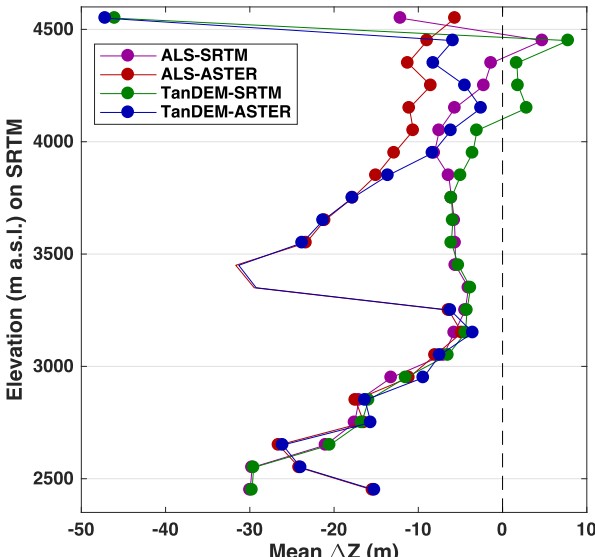

**Figure 9.** Vertical profile of measured ice elevation change on Universidad Glacier. All ΔZ measurements were averaged per 100 m elevation bin. Only GA pixels were used.

## 5. Discussion

### 5.1. TanDEM-X DEM Uncertainty

As stated before, the TanDEM is a product made by averaging of several intermediate DEMs (iDEM). The final TanDEM-X DEM is a weighted average of these iDEMs, with weights depending on their quality assessed with the standard deviation of height error over each individual iDEM [23]. The X-band radar wave used by TanDEM-X satellites has been shown to penetrate dry snow by up to 6 m, but the penetration of wet snow is smaller than 10 cm [46]. This difference in radar backscatter for the different surface types could contribute to the elevation error when DEMs from different seasons are combined. In our case however, the factors of multi-temporal averaging and variable radar penetration had only a limited impact on the performance of TanDEM, contributing less than 0.5 m of bias relative to ALS both over stable areas (Table 2) and over the glacier (Table 3). The reason for this may be the limited contribution of winter iDEMs (Figure 1B), when radar penetration is high, to the averaged product.

The majority of iDEMs contributing to the DEM in our study area represent spring, summer and early autumn with winter being the most underrepresented season with only 17 iDEMs versus 20 in autumn, 31 in spring and 34 in summer (Figure 1B). These numbers already show the small contribution of winter values to the final DEM. Furthermore, it is possible that during winter acquisitions radar penetration of the snow cover introduced enough uncertainty (reflected as large standard deviation of height errors) to force the assignment of lower weights of these iDEMs, compared to summer acquisitions, further suppressing their influence on the final output. In that case, the resulting average surface would lie even closer to the actual summer glacier surface than if the acquisition weights from the four seasons were considered equal.

It could be argued that immediately after a summer snowfall the snow would be dry and permeable to the radar signal, influencing even the summer acquisitions. We assume that the snow quickly transitions to a wet state in the conditions of Andean summer. In addition, the summer precipitation in this area is low, so this situation is not frequent enough to influence the TanDEM-X DEM. The early spring and late autumn snowfalls could potentially contribute error in the same way as winter snowfall would do, when temperatures are lower than in summer. The number of acquisitions during these months (April, May, and September) is however low, similar to the numbers from winter

months (Figure 1B). Thus, we assume that the dry snow in spring/autumn has a comparable, low, influence on the final TanDEM-X DEM as the winter acquisitions described above.

Hence, the two types of surface influencing the most the final elevation of TanDEM over SA would be that of bare rocks and wet snowpack in spring. It can be assumed that dry snow is only present on Universidad Glacier during winter months when air temperatures are below freezing and possibly at night when the surface snow refreezes. This can be supported by calculations of the glacier surface energy balance between November 2012 and April 2014 by Kinnard et al. [27]. During winter only occasional melting was observed, with most of energy dissipated radiatively from the surface of the glacier.

Nonetheless, the glacier elevation measured in springtime with X-band radar would be influenced by wet snow. The result could be an overestimation of $Z$ due to the contribution of transient snow cover thickness to the elevation signal. Such effect, however, was not observed in our results, with mean $\Delta Z$ only 0.04 m below 0 on GA (Table 3) and overall positive bias on SA (Table 2 and Figure 5). This testifies to the quality of iDEM averaging algorithm, which managed to reduce the effect of snow penetration.

The more important source of error in TanDEM is the bias caused by terrain irregularities. The accuracy of the $Z$ values deteriorates significantly for slope higher than 40 degrees. The same tendency, also with a threshold of 40°, was noted by Dehecq et al. [46], where DEMs were built by the authors from TanDEM-X SAR acquisitions. There are erroneous areas in TanDEM along the western margin of the glacier (background of Figure 1B) and their effect on $\Delta Z$ is depicted on Figures 3D,E and 4D,E. Such strong artifacts limit the use of this dataset over steep and rugged terrain.

The reason for the artifact's spatial distribution may be related to the geometry of SAR scanning by TanDEM-X satellites. Eineder [42] proposed a method of assessing the set of slopes reliably visible to a spaceborne radar scanner. He draws a slope/aspect polar plot where the range of slopes useful for InSAR analyses can be depicted, knowing the incidence angle of the radar beam and the angle of satellite orbit. Such analysis does not account for shadowing of areas by the terrain [42], so some areas can lie in radar shadow, despite being in the useful slope range. We performed a similar assessment for one of the iDEMs overlapping with our study area. The range of incidence angles in the set of iDEMs is 31.32°–46.83°. We have chosen the lowest of them for the purpose of visualization of useful slope range, because it gives the largest set of slopes, which overlaps the sets produced by higher incidence angles.

The inclination of the orbit of TanDEM-X satellites is 97.4° [47]. We have plotted usability plots of two satellite passes over the terrain (ascending and descending). The analysis was overlain on a set of slope/aspect pairs derived from 12 m ALS and slope limits enclosing 90%, 99% and 99.9% of all pixels (Figure 10). The result shows that coverage of our study area by the TanDEM is below 90%. Most pixels unusable for InSAR lies in the western and eastern sections of the plot, corresponding to western and eastern aspects of the terrain. This overlaps with the high-error areas visible on Figure 1D. The usable set shown on Figure 10 is an upper bound estimate.

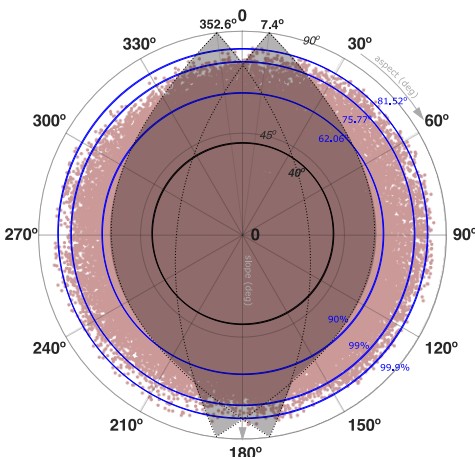

**Figure 10.** Polar plot of slope/aspect pairs usable for a single InSAR acquisition (single iDEM) with incidence angle of 31.32° over our study area. Red points indicate pixels of ALS, blue circles—slope values corresponding to 90%, 99% and 99.9% of all pixels and shaded area—range of usable slopes obtained with the method of Eineder [42].

The last source of possible error in Z is connected to the ALS DEM. The original product had a very high resolution of 1 m, which had to be degraded to 12 and 30 m to match the other datasets. The downsampling was done by means of averaging, which could introduce errors. The aggregation error was tested by interpolating the 30 and 12 m ALS DEMs back to 1 m with bilinear interpolation and calculating mean ΔZ between the original and the interpolated DEMs. The results for the SA were 0.0057 m and 0.0574 m and for GA 0.0029 m and −0.0467 m for the 12 and 30 m DEMs, respectively. Thus, these numbers indicate that the interpolation of ALS resulted in negligible Z error to the final results.

### 5.2. TanDEM-X DEM Performance

Results listed in the last columns of Table 2 show that in our study area both TanDEM12 and TanDEM30 have only partially met the HRTI-3 and DTED-3 goals of vertical accuracy. We compare the results to the relative vertical accuracy requirements [19] as our study area is smaller than 1° × 1°. The LE90 of TandDEM12 (1.09 m) is below the stated limit (2 m) when pixels with slope <11.31° (20%) are considered but exceeds the target (16.50 vs. 4 m) for slopes between 11.31° and 40°. Similarly, both TanDEM30 and TanDEM30bili are well below the goal (15 m) for $\alpha$ < 11.31° (2.03 m and 1.35 m respectively), while for the higher slopes interval the results are above the required 15 m (26.77 m and 22.73 m). All three TanDEMs exhibit very high LE90 for slopes >40°. These pixels were the most affected by errors and outliers and were therefore filtered out during the analyses. LE90 larger than 200 m reinforce the need to remove these pixels before carrying any analyses on the DEMs. Although this poor performance over steep slopes may seem to limit the use of the TanDEM-X DEM, the area over which these results were acquired is small, corresponding to a single glacier basin and several mountain peaks surrounding it. Thus, the inaccuracies we have calculated should be interpreted locally and with careful application to the TanDEM-X DEM product as a whole or even in studies at a regional scale.

In our opinion, TanDEM-X DEM should be inspected closely before any work related to mountainous areas is conducted with this product. There is hope, however, that these problems will be mitigated in future as more acquisitions with different viewing angles will become available [22]. In this work the impact of the slope-related artifacts on the glaciological results is small. The aspect-slope histogram of Universidad Glacier mainly overlaps with the section of the SA aspect-slope error heatmap which is not affected by this problem (Figures 3H and 4H). In other words, there are not enough steep pixels on the glacier to produce a significant difference between the corrected and uncorrected data.

Nevertheless, the ALS-based multi-temporal mean $\Delta Z$ are different from TanDEM-based ones by a larger amount than the mean $\Delta Z$ between ALS and TanDEM (Table 3). The areas of the highest ALS-TanDEM $\Delta Z$ are located on the western margins of the western accumulation basin where ALS-SRTM and TanDEM-SRTM differences are the highest as well. The ALS-TanDEM difference is negative ($\Delta Z < 0$) in this area (Figure 5A), indicating that the TanDEM surface lies higher than ALS. In fact, it is high enough to lie even above the ASTER/SRTM surfaces, contributing positive $\Delta Z$ to the distribution. This way, the right tail of GA $\Delta Z$ distribution is heavier in the TanDEM change detection cases than in the ALS ones (Figures 6 and 7) and shifts the mean $\Delta Z$ towards 0. This is also seen in the altitudinal distribution of elevation change (Figure 9), where $\Delta Z > 0$ is seen in 4 altitude bands when TanDEM-SRTM map is analyzed and only in a single one in the ALS-SRTM case.

The sudden drop in the mean $\Delta Z$ in the highest altitude band (Figure 9) corresponds to the low-$\Delta Z$ area in the NW part of the western basin, above 4500 m a.s.l., seen on both ALS-SRTM and TanDEM-SRTM maps. This part of the glacier lies very close to the valley slopes and is subject to the TanDEM high-slope artifacts (Figure 1D). As GA was not filtered to remove $\alpha > 40°$ pixels, the problematic areas could impact the results this way. The presence of $\Delta Z < 0$ in this area in all difference maps (Figures 6 and 7) suggests that it is caused by thinning in the highest part of the glacier, not errors in the datasets. On the other hand, overestimation of the thinning in this area by TanDEM-based maps, in an area prone to systematic errors, points to ALS as the more accurate representation of Universidad Glacier elevation in 2013 than TanDEM. Imperfections of SRTM in the same area (Figure 2C) explain the mean $\Delta Z$ drop in ALS-SRTM in the highest altitudes, but the effect is much smaller, than in case of TanDEM.

Our results confirm that the previously reported high accuracy of TanDEM over land surfaces also applies to glacier surfaces. A global study of TanDEM accuracy by Wessel et al. [24] has shown the product to have $\sigma$ between 0.95 and 2.27 m as compared to GPS measurements over a wide range of land cover types. The same study reported a mean bias below 0.2 m for TanDEM12. Our calculated inaccuracies are higher, with a mean SA bias after corrections of 0.62 m (Figure 5) and $\sigma_c = 3.48$. The Wessel et al. [24] study, however, focused on relatively flat areas (majority of control points with $\alpha < 10°$) while our area is in mountainous terrain. In regional comparisons with ALS-derived DEMs, areas with slope $>30°$ have been shown to have significantly higher $\sigma$ than flat areas [24], in line with our results. Similar results were reported by Grohmann [25], who compared TanDEM-X DEM with several other satellite-derived DEMs. In this case, their reported $\sigma$ was higher than those reported by Wessel et al. [24], owing to the higher uncertainty of the reference satellite products resulting from peculiarities of the sensing and production procedures. In comparison of several global DEMs to differential GPS measurements by Purinton and Bookhagen [26] TanDEM12 also showed good performance for hydrological and geomorphometric tasks. The standard deviation of $\Delta Z$ relative to GPS was used as the measure of accuracy. The $\sigma$ achieved in the study were lower than ours (TanDEM12: 1.97 m, TanDEM30: 2.42 m, ASTER GDEM: 9.48 m, SRTM: 3.33 m), but their study focused on areas with gentler slopes ($\alpha < 30°$). In this type of terrain, the vertical accuracy of TanDEM12 was determined to be better than 2 m. Purinton and Bookhagen [26] drew conclusions similar to ours regarding the comparison of different global, satellite DEMs. ASTER GDEM was shown to be noisier than radar-based SRTM and TanDEM-X DEMs with single-scene ASTER generated by the authors from image stereopairs exhibiting even lower quality.

## 5.3. ASTER DEM Quality

The divergence of results obtained from ASTER and SRTM point to problems with one of the DEMs. Visual analysis of the maps on Figures 6 and 7 led us to conclude that ASTER is a less reliable source to reconstruct past glacier elevation than SRTM. We interpret the large red areas seen on the ASTER-derived $\Delta Z$ maps in the higher part of the glacier to be a result of processing errors at the stage of batch DEM creation from stereoimagery. The method of DEM production from ASTER imagery relies on the calculation of parallax difference between nadir-looking and angled ASTER

images. This difference is computed based on the cross-correlation of small chips of one image versus the other [48]. For good results, however, the algorithm requires characteristic tie-points which can be matched on both images. The often uniform and bright surface of glaciers, especially in their accumulation area, can provide very few such features, leading to imperfections in the final product.

Indeed, the most suspicious parts seen in Figure 7 are located in the higher parts of the glacier, in the accumulation area where albedo is higher and contrast between different surface types is lower [27], while the tongue or icefalls which have lower albedo and increased roughness offer better contrasts and are hence represented correctly. In the accumulation area the probability of having fresh, bright snow is much higher than in lower parts of the glacier where moraines, ogives, crevasses, debris, and rough ice provide a diversity of surfaces allowing for successful stereomatching. An additional inspection of ASTER has shown discontinuities in the problematic areas, despite clouds being absent on the source multispectral images (Figure 2). Additionally, small artifacts have been found in ASTER even in low-altitude area, such as at the glacier's tongue in a section where contrasting patterns of debris and clear ice offer a clear contrast. These artifacts impact the altitudinal distribution of mean $\Delta Z$, where ASTER-based maps diverge strongly from SRTM-based maps below 2600 m a.s.l. and between 3300 and 3900 m a.s.l. (Figure 9). The higher reliability of SRTM over ASTER for glacial application is also seen in their $\sigma$ values, which is slightly lower for SRTM (Figure 6) than for ASTER (Figure 7). From these observations and results we conclude that SRTM is a more reliable source for the past state of the glacier than ASTER and further discuss the reconstructed glacier mass balance based on the ALS-SRTM 12 m results only. Albeit SRTM shows imperfections, the affected area is small and limited to a particular section of the glacier, while ASTER's artifacts are spread across many elevation bands and cover large area. The ASTER discontinuities in the lower part of the glacier are spatially limited and ASTER could this potentially still be used to track changes of Universidad Glacier in the ablation area only.

### 5.4. Glacier Change Detection

The large magnitude of thinning in the lowest part of the glacier points to a retreat trend, in line with the negative mean $\Delta Z$. According to Barcaza et al. [49], Universidad Glacier lost 0.17 km$^2$ of surface area between 2000 and 2015, hence experiencing a frontal retreat. The thinning rate is not uniform over the lower tongue, where alternating bands of higher and lower $\Delta Z$ are present (Figure 6). These regions can be related to debris-covered sections of the tongue and elevated bands seen on hill-shaded DEMs (Figures 1C and 2A). A thin debris cover is known to enhance ablation while a thicker cover insulates the ice and slows down ablation [50]. The debris cover on the tongue and lateral moraines seems to be thick enough to hinder the correct detection of the glacier extent (as mentioned in Section 3.2). The variable pattern of thinning rate along a transect perpendicular to the ice flow in the tongue section (Figure 8) fits the sections of alternating clean ice and debris in the area. This reveals that the debris cover is sufficiently thick to insulate the underlying ice and decrease surface melt. The result is the formation of clean-ice troughs between debris-covered ridges.

Ice thickening can be observed in the topmost areas of the accumulation zone as well as in a restricted region directly above the major icefall in the western basin. We interpret the positive geodetic $\Delta Z$ seen in the upper part of the western basin to result from increased accumulation in this area. Still, the two accumulation basins of the glacier are losing ice over most of their areas, albeit at a slower rate than the tongue. On the other hand, the positive vertical change above the icefalls is likely driven by local compressive ice flow, which pushes the ice upwards, rather than by positive surface mass balance due to accumulation of firn and snow. The ice flow velocity was shown to be highest beneath the ice fall areas [51], supporting this claim.

The ice thickness changes (Figure 6) and calculated geodetic mass balance (Table 3) of Universidad Glacier can be compared to previously published measurements of glaciological surface mass balance. An ablation stake study [27] has shown that at higher altitudes (3500–4200 m a.s.l.) the surface mass balance was positive in the hydrological year 2012–2013 (up to +2 m w.e.a$^{-1}$), but negative in 2013–2014

($-4$–0 m w.e.a$^{-1}$); however the stakes used for this study were absent in areas of $Z$ increase in the highest parts of the basins. Additionally, Bravo et al. [29] modeled melt of Universidad Glacier with a temperature index model tuned to field measurements in the ablation zone. They concluded that ice melt provided over 80% of discharge to the proglacial river, with the rest attributed to snowmelt. Simulated melt rates for the 2009–2010 ablation season ranged from 1 m w.e. in the accumulation area to 10 m w.e. in the lower part of the tongue. Our result of 0–1 m w.e.a$^{-1}$ ($-10$–0 m w.e.a$^{-1}$ $\Delta Z$ over 2000–2013) of ice elevation loss rate in the accumulation area are in agreement with the simulated melt rate from Bravo et al. [29] and observations from the 2013/14 year. Consistently negative geodetic $\Delta Z$ may stem from out-of-balance vertical component of ice flow, a factor not accounted by ablation stake measurements or melt rate modeling. The positive measured point mass balance and negative long-term elevation change seen in the accumulation area may mean that the amount of ice flowing down-glacier from the basins is not compensated for by the new accumulation. A similar pattern can be observed in the ablation zone where emergent ice flow is not high enough to compensate for ice loss due to extremely high melt, reaching 8-11 m w.e. in the terminal part of the glacier in 2009–2010 season [29]. Kinnard et al. [27] showed that the surface mass balance at the ablation stakes at low to mid altitudes was $-5$ m w.e.a$^{-1}$ and $-10$ m w.e.a$^{-1}$ in 2012–2013 and between $-6$ m w.e.a$^{-1}$ and $-14$ m w.e.a$^{-1}$ in 2013–2014 hydrological years. The geodetic $\Delta Z$ exceeds $-40$ m at some points on the lowermost part of the tongue (more than 3 m w.e.a$^{-1}$ of ice loss). The later result, however, includes vertical, emergent, movement of ice caused by compressive flow at the terminus, which adds a positive $\Delta Z$ signal to the final result. The ablation rate, however, is too high to be compensated by the emergence of ice, thus leading to the observed surface lowering and frontal retreat that follows climatic forcing [51].

During the 2012–2013 hydrological year the measured glaciological mass balance of Universidad Glacier was a loss of $-0.32 \pm 0.40$ m w.e. [27], which is smaller than the geodetically reconstructed long-term loss of $-0.44 \pm 0.08$ m w.e.a$^{-1}$. However, the measured mass balance was found to be drastically more negative the following year (2013–2014: $-2.53 \pm 0.57$ m w.e.a$^{-1}$), due to much reduced snowfall, lower albedo and a lower cloud cover compared with the preceding year [27]. It is, therefore, hard to judge based on these limited data whether the general 2000–2013 thinning trend is slowing, as suggested by the 2012–2013 observations, or if the 2013–2014 year is more representative of the general climatic trend leading to increased ablation. In any case, in light of the recent climate projections which show warming and large reduction in winter precipitation (50%) in central Chile by the end of the 21st century [52,53], and the strong positive feedback of reduced snow cover on ablation observed by Kinnard et al. [27], the table seems to be set for increasingly negative mass-balance conditions on Universidad Glacier in the future.

## 6. Conclusions

TanDEM-X DEM is a high-quality DEM which, however, requires several preprocessing steps to be used reliably for the analysis of alpine glaciers. Filtering of outliers and areas with slopes steeper than $40°$ is the most important of them, as steep surfaces are often represented erroneously and contribute large error to the data. Although TanDEM depicts a multi-year average of ice elevation, in our case it approximated well the glacier surface in April 2013, which was close to the midpoint of the intermediate DEM (iDEMs) acquisition period, and with an accuracy sufficient for change detection studies. The impact of penetration of the radar signal into dry snow on the results is likely very small in our study due to small number of winter iDEMs included in the final product over Universidad Glacier, and the fact that wet snow is found for a large portion of the year on the glacier. These two factors should, however, be investigated when glaciological studies are undertaken with TanDEM-X DEM in other regions of the world, as the iDEM distribution and dry snow frequency could vary significantly between regions.

The median bias of the TanDEM-X DEM over the non-glacierized area was $0.02 \pm 3.48$ m for the 12 m delivered product and $-0.08 \pm 7.57$ m for the 30 m product. The bias of both datasets is

lower than that of ASTER and SRTMv3 DEMs, which are much-used products in glaciological research. The precision ($\sigma_c$) of TanDEM12 was better than that of ASTER and SRTM, while that of TanDEM30 was comparable to the other 30 m DEMs. The calculated LE90 scores indicate that TanDEM12 did not meet the targeted HRTI-3 standard, but our study was conducted in a small region over the steep mountainous terrain of the Chilean Andes, thus the result might not be representative for TanDEM-X DEM as a whole or at a regional scale. They do call however for careful accuracy assessment over this type of terrain in future studies.

The uneven performance of TanDEM12 and TanDEM30 over both rocky and glacierized terrains indicate that the former should preferably be used for Earth surface analyses. As the resampling of SRTM to coarser 12 m introduced only a very small error it is possible to employ the TanDEM12 in tandem with older, coarser, datasets thus retaining its major advantage even when working with datasets with originally lower resolutions.

The stereo-optical ASTER DEM overestimated the elevation of the glacier, possibly as a result of a lack of tie-points over the low-contrast accumulation area of the glacier, which confuses the stereomatching algorithms. This results in overestimated ice loss values, compared to using the SRTM DEM derived from radar altimetry as the historical baseline for geodetic mass balance calculations.

Using the SRTM DEM as a reference historical surface (2000) and ALS as the more contemporary image (2013) we report that Universidad Glacier has thinned by $6.77 \pm 0.34$ m on average over 13 years, which corresponds to a yearly ice loss rate of $-0.44 \pm 0.08$ m w.e.a$^{-1}$. Thinning was observed across the whole glacier, but with largest thinning rates observed in the lowest part of the glacier, pointing to a retreat trend of the glacier tongue. Debris cover in this area is sufficiently thick to impact ablation patterns and promote the formation of debris-covered ridges. Mass loss was also generalized in the accumulation area, albeit at slower rates than on the tongue. Some thickening was found at the highest section of the glacier show, perhaps resulting from increased accumulation, or localized avalanching. Localized bulging was also observed above the icefall which may have resulted from ice flow dynamics, such as ice uplifting by compression above the icefalls.

**Author Contributions:** J.P. performed most of the calculations, interpretations and wrote the paper. M.P. designed the study. M.P. and C.K. contributed to calculations and interpretation of results as well as to the writing of the article. C.K. and R.U. provided aerial laser scanning data.

**Funding:** J.P. is funded by Institute of Geophysics of Polish Academy of Sciences (IGF PAS) statutory funds and funds of the Leading National Research Centre (KNOW) received by the Centre for Polar Studies for the period 2014-2018. C.K. is funded by the Canada Research Chair program and the Natural Sciences and Engineering Research Council of Canada. M.P. was supported by CONICYT-FONDECYT Iniciación grant #11170937 and by CECs, which is funded by the Base Finance programme of CONICYT, Chile. R.U. thanks the CONICYT/FONDAP/15130015 Project. Aerial laser scanning and its preprocessing was funded by Fondo de Innovación para la Competitividad del Ministerio de Economía, Fomento y Turismo, Gobierno de Chile (R.U., C.K.). The APC was funded by the statutory funds of IGF PAS. TanDEM-X DEM data was provided by the German Aerospace Center (DLR) through an Announcement of Opportunity and Proposal Call (proposal DEM_GLAC1874).

**Acknowledgments:** TanDEM-X DEM data © DLR 2017. SRTM DEM, the ASTER DEM V003, ASTER and MODIS Cloud Mask were retrieved from the online EarthData portal, courtesy of the NASA EOSDIS Land Processes Distributed Active Archive Center (LP DAAC), USGS/Earth Resources Observation and Science (EROS) Center, Sioux Falls, South Dakota, https://earthdata.nasa.gov/. ASTER DEM V003 (ASTER) is a product of METI and NASA. The authors thank Dr Ian S. Evans and an anonymous reviewer for comments which led to improvement of this work.

**Conflicts of Interest:** The authors declare no conflict of interest. The funding sponsors had no role in the design of the study; in the collection, analyses, or interpretation of data; in the writing of the manuscript, and in the decision to publish the results.

## Abbreviations

| | |
|---|---|
| ALS | Aerial Laser Scanning DEM |
| ASTER | ASTER DEM V003 |
| ASTER12 | ASTER DEM V003 interpolated to 12 m resolution |
| $\alpha$ | surface slope |
| DEM | Digital Elevation Model |
| DLR | German Aerospace Center |
| DTED-2 | Digital Terrain Elevation Data—2 standard |
| $\Delta x$ | spatial resolution of a dataset |
| $\Delta Z$ | difference of elevation |
| GA | glacier area |
| GC | geometric correction |
| HRTI-3 | High-Resolution Terrain Information 3 standard |
| iDEM | intermediate DEM |
| JAXA | Japan Aerospace Exploration Agency |
| L | surface area of area of interest |
| LiDAR | Light Detection and Ranging |
| m w.e. | meters of water equivalent |
| $n$ | nugget of semivariogram |
| NASA | National Aeronautics and Space Administration |
| $\Psi$ | aspect |
| $r$ | range of semivariogram |
| RGI | Randolph Glacier Inventory |
| $s$ | sill of semivariogram |
| SA | stable area |
| SAHC | Slope-aspect heatmap correction |
| SAR | synthetic aperture radar |
| $\sigma_c$ | standard deviation |
| $\sigma_p$ | uncertainty of the mean of partially spatially correlated dataset |
| $\sigma_u$ | uncertainty of the mean of spatially uncorrelated dataset |
| SRTM | SRTMv3 DEM |
| SRTM12 | SRTMv3 DEM interpolated to 12 m resolution |
| TanDEM12 | TanDEM-X DEM 12 m |
| TanDEM30 | TanDEM-X DEM 30 m |
| TanDEM30bili | TanDEM-X DEM 30 m created by bilinear interpolation of TanDEM12 |
| Z | elevation |

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
