# Peer review of "Performance Assessment of TanDEM-X DEM for Mountain Glacier Elevation Change Detection"

_remotesensing, doi:10.3390/rs11020187_

Round 1
Reviewer 1 Report
Comments by Dr. Ian S. Evans, Durham University, on mdpi Remote Sensing 394988:
Performance Assessment of TanDEM-X DEM for Mountain Glacier Elevation Change Detection. Julian Podgórski, Christophe Kinnard, Michał Pętlicki and Roberto Urrutia
This submission presents detailed comparisons of TanDEM-X, SRTM and ASTER DEMS (which are available almost universally) with an airborne laser-scanned DEM (ALS), for a Chilean glacier and its surrounds. It is well written and succinctly expressed, with concise illustrations rich in detail. Methods, results and interpretations are thoroughly covered. All Figures are relevant and useful, and the literature is very well covered.
From the glacier change point of view, this contributes a new, highly accurate ALS for comparison with the older SRTM DEM, and possibly the ASTER DEM, providing integrated glacier surface and volume change over 2000-2013. A second theme is the comparison of TanDEM-X, averaged over 3 years, with the ALS. As the ALS is more accurate, comparison of TanDEM-X with SRTM and ASTER is largely redundant for change of this glacier, but of interest in supporting use of TanDEM-X with SRTM elsewhere, where ALS is unavailable. Interweaving of the two themes does cause problems of structure, but the accuracy comparisons (and spatial error patterns) interact strongly with the glacier change study so it is useful to have both in one paper.
ASTER is correctly discarded as inaccurate over glacier accumulation areas. However, it could be used for ablation areas, and comparison with ALS there would improve the record of change over time. It is worth inserting a sentence on the ASTER problems in the abstract.
The inaccuracy of TanDEM-X on steep slopes is important and deserves fuller discussion (as to possible explanations).
Traditional studies of glacier mass balance plot change (and inputs and outputs) against altitude. That could be done here for altitude change by taking the results from Fig. 5A (& B) and averaging them for each (initial) altitude band..
Accuracy is described here as higher when error measures are higher: this must be avoided. E.g. ‘accuracy … is below 0.5 m’ in the abstract. Either discuss inaccuracy being higher/lower, or error being higher/lower.
There are a number of corrections needed, either editing errors or polishing of expression (mainly definite and indefinite articles), possible inconsistencies, and one possible small change in structure:
DETAILS:
line 10 ‘is better than 0.5 m …’
38 ‘often a preferred’
45 ‘the PRISM’
57-60 It appears from what follows, and it is implied here, that individual iDEMs were not used, presumably because they were not available. As they might have been useful for studies of 2011-2014 glacier change (some seasonal function could have been fitted in combination with a linear trend), it should be stated explicitly that iDEMs were not used (and not available?).
60 2011-2014 is not a ‘long period’! I suggest ‘even over a 3-year period, and seasonally’
72 130 km surprised me, as the dot in Fig. 1A seems somewhat misplaced: according to the scale there, it is plotted just 82 km south of Santiago.
76 ‘supply: glacial’
77 ‘there’ rather than ‘for them’
89 and 252; 90, 132 and 264: use of ‘natively’ and ‘native’ here surprised me. Usually ‘native’ is applied to persons, or ‘not introduced’ or ‘occurring in a pure state in nature’ (New Webster’s Dictionary). Would ‘initial’ or ‘original’ or ‘as supplied’ be more appropriate?
138 ‘the glacierized’
145-146 and Fig.1 Please explain why the glacier outline in C omits some glaciers outlined in D, e..g. in SW and NE: are these glaciers used or of any relevance?
Fig. 1B caption: maybe ‘temporal distribution of SAR’
Fig. 1B the ‘clouded areas’ dots in A are black – not white as in key.
171 Did you ask TanDEM-X admin about the weights? Did they refuse? It seems a strange detail to be secretive about.
198 I thought GC was a correction ‘based on slope-aspect’. The distinction between two slope correction methods was not clear in section 3.2: please clarify the distinction between GC and the slope-aspect correction.
235 delete ‘the’
246 ‘on our quality …’
Fig.2 The scale key for E gives ΔZ near the mid-points of colour classes, but F gives them near class boundaries: could both be labelled at the six class boundaries? Also in Fig.3. And could E, F and G have the SAME classes, to permit visual comparison of these graphs? Finally, the five (or six)colours need to be more distinct: the bottom three are hardly distinguishable.
Figs. 2 & 3 The initial sentence in each misleadingly implies that all 8 figures are based on SA. Therefore I suggest a change to ‘on SA (A-E), and applied to GA (F-H)’
277 replace ‘Indeed’ with ‘Also’ - I do not see the connection between the first sentence and the second.
279-280 ‘lower’ refers to aspect – as this has not previously been discussed, some mention as to why aspect is relevant would be welcome. (It is clear from Fig. 3H that aspects around north are uncommon on the glacier, which faces mainly SSE).
293 due to shifts in the class boundaries between colours, it is not clear that Figs. 2 & 3 support this statement.
298-300 reference to Figs. 5 & 6 is desirable here.
301 ‘DEMs’
316 ‘The case’
Section 4.2 I do not accept that comparing ALS with TanDEM-X (2013-2013) gives any information on glacier mass balance (as the heading would imply). ALS-TanDEM=X differences on the glacier depend mainly on the (unknown) weights applied to different TanDEM-X iDEM dates. Perhaps most of lines 293-296, 304-319 and Fig.4 should be given a separate section. All reference to SA is relevant to general quality assessment, not directly to glacier balance, so fuller separation from discussion of the latter would improve clarity.
Table 3 Insert ‘for GA’ ?
Table 3 σp appears here, but neither σc nor σu do. Figs. 4-6 give σc and σ , but is σu nowhere.
Fig. 5B You might explain why no TanDEM for SA is included here, whereas it was previously – in Fig.4.
378 While a wet surface might be expected for much of the summer, I would expect dry snow after fresh, cold snowfalls at high altitude in parts of spring and autumn.
391 ‘dataset’
437 EITHER ‘accuracies are poorer’ OR ‘inaccuracies are higher’.
445 ‘by Puriton …’
446 ‘as the measure’
449 NOT ‘accuracy … below’ but ‘accuracy better than 2 m’
456 ‘Figure 6 ‘ as ASTER is not in Fig.4
473 I would like these discontinuities to be illustrated.
484-485 How does Fig.5 show this? It depends how the dark red stripes near the terminus line up with the medial moraine stripes in Fig. 1D. Cross-sections of this part of each Figure would help support (or refute, therefore drop) this statement.
512 ‘A similar pattern …’
523 ‘was a loss of …’
524 ‘loss’ rather than rate.
527 delete ‘either’
547 ‘over the non-glacierized …’ [note; much of the rocky area has been glaciated]
619 delete ‘<’ ?
620 ‘correlated’
621 ‘uncorrelated’
- Ian S. Evans 26 November 2018
Author Response
REVIEW 1
Open Review
( ) I would not like to sign my review report
(x) I would like to sign my review report
English language and style
( ) Extensive editing of English language and style required
( ) Moderate English changes required
(x) English language and style are fine/minor spell check required
( ) I don't feel qualified to judge about the English language and style
Yes | Can be improved | Must be improved | Not applicable | |
Does the introduction provide sufficient background and include all relevant references? | (x) | ( ) | ( ) | ( ) |
Is the research design appropriate? | (x) | ( ) | ( ) | ( ) |
Are the methods adequately described? | (x) | ( ) | ( ) | ( ) |
Are the results clearly presented? | (x) | ( ) | ( ) | ( ) |
Are the conclusions supported by the results? | (x) | ( ) | ( ) | ( ) |
Comments and Suggestions for Authors
Comments by Dr. Ian S. Evans, Durham University, on mdpi Remote Sensing 394988:
Performance Assessment of TanDEM-X DEM for Mountain Glacier Elevation Change Detection. Julian Podgórski, Christophe Kinnard, Michał Pętlicki and Roberto Urrutia
This submission presents detailed comparisons of TanDEM-X, SRTM and ASTER DEMS (which are available almost universally) with an airborne laser-scanned DEM (ALS), for a Chilean glacier and its surrounds. It is well written and succinctly expressed, with concise illustrations rich in detail. Methods, results and interpretations are thoroughly covered. All Figures are relevant and useful, and the literature is very well covered.
From the glacier change point of view, this contributes a new, highly accurate ALS for comparison with the older SRTM DEM, and possibly the ASTER DEM, providing integrated glacier surface and volume change over 2000-2013. A second theme is the comparison of TanDEM-X, averaged over 3 years, with the ALS. As the ALS is more accurate, comparison of TanDEM-X with SRTM and ASTER is largely redundant for change of this glacier, but of interest in supporting use of TanDEM-X with SRTM elsewhere, where ALS is unavailable. Interweaving of the two themes does cause problems of structure, but the accuracy comparisons (and spatial error patterns) interact strongly with the glacier change study so it is useful to have both in one paper.
ASTER is correctly discarded as inaccurate over glacier accumulation areas. However, it could be used for ablation areas, and comparison with ALS there would improve the record of change over time.
It was shown on Figure 2 and accompanying discussion that DEM errors are present in ASTER on the glacier’s tongue as well, (L544-550 An additional inspection of ASTER has shown discontinuities in the problematic areas, despite clouds being absent on the sourcemultispectral images (Fig. 2). Additionally, small artifacts have been found in ASTER even in low-altitude area, such as at the glacier’s tongue in a section where contrasting patterns of debris and clear ice offer a clear contrast. These artifacts impact the altitudinal distribution of mean DZ, where ASTER-based maps diverge strongly from SRTM-based maps below 2600 m a.s.l. and between 3300 and 3900 m a.s.l. (Fig. 8).) The potential of ASTER for ablation-area analyses has been highlighted as well (L556-558 The ASTER discontinuities in the lower part of the glacier are spatially limited and ASTER could this potentially still be used to track changes of Universidad Glacier in the ablation area only.)
It is worth inserting a sentence on the ASTER problems in the abstract.
A sentence “The ASTER DEM contains errors, possibly resulting from imperfect DEM creation from stereopairs over uniform ice surface.” was introduced to the abstract (L16-17)
The inaccuracy of TanDEM-X on steep slopes is important and deserves fuller discussion (as to possible explanations).
To test a hypothesis, that geometry of SAR scanning impacted the final quality of DEM a method by Eineder (2003) was used to determine sections of terrain, that can be effectively mapped by InSAR. Fig. 8 shows the result of this exercise and confirms, that our study area is not completely covered in satisfactory manner by the radar acquisitions, that were used to create iDEMs. L420-433 state the hypothesis and discuss the test and its results:
Traditional studies of glacier mass balance plot change (and inputs and outputs) against altitude. That could be done here for altitude change by taking the results from Fig. 5A (& B) and averaging them for each (initial) altitude band..
A plot of mean elevation change against SRTM elevation was added to the manuscript (Fig. 8) and discussed (L372-378 Summary of mean DZ change with altitude shows that the glacier has thinned in all altitude bands except one (Fig. 8). The largest elevation change is observed at the lowest altitudes (glacier terminus). A steep decrease in absolute DZ is observed between 3100 and 4000 m a.s.l, corresponding to the tongue and icefalls. The highest part of the glacier has the lowest absolute elevation changes. The direction of change differs depending on whether TanDEM or ALS is used as the later DEM. TanDEM-SRTM shows thickening of the glacier, while ALS-SRTM DZ is positive in only a single elevation band., 493-500 This is also seen in the altitudinal distribution of elevation change (Fig. 8), where DZ>0 is seen in 4 altitude bands when TanDEM-SRTM map is analysed and only in a single one in the ALS-SRTM case. The sudden drop in the mean DZ in the highest altitude band (Fig. 8) corresponds to the low-DZ area in the NW part of the western basin, above 4500 m a.s.l., seen on both ALS-SRTM and TanDEM-SRTM maps. This part of the glacier lies very close to the valley slopes and is subject to the TanDEM high-slope artifacts (Fig. 1D). As GA was not filtered to remove a>40o pixels, the problematic areas could impact the results this way., 548-550 These artifacts impact the altitudinal distribution of mean DZ, where ASTER-based maps diverge strongly from SRTM-based maps below 2600 m a.s.l. and between 3300 and 3900 m a.s.l. (Fig. 8).). The figure has shown, that mean elevation change is below 0 in nearly all altitude bands, strengthened the argument against using ASTER DEM for mass balance assessment and revealed influence of slope error of TanDEM-X DEM on some parts of the glacier, aside of the surrounding slopes.
Accuracy is described here as higher when error measures are higher: this must be avoided. E.g. ‘accuracy … is below 0.5 m’ in the abstract. Either discuss inaccuracy being higher/lower, or error being higher/lower.
We have edited mentions of accuracy and error to be expressed correctly (e.g. L104 vertical inaccuracy lower than 0.1 m., L276 the error of all DEMs is below 0.5 m,, L4510-511 Our calculated inaccuracies are higher)
There are a number of corrections needed, either editing errors or polishing of expression (mainly definite and indefinite articles), possible inconsistencies, and one possible small change in structure:
DETAILS:
line 10 ‘is better than 0.5 m …’
The general statement of accuracy has been replaced with exact values (L12-13 The vertical accuracy (+/- standard deviation) of TanDEM-X DEM over non-glacierized area is very good at 0.02+/-3.48 m…, L15-16 …accuracy, -0.08+/-7.57 m, is better, than…)
38 ‘often a preferred’
The sentence was corrected as suggested (L42-43 …radar altimetry is often a preferred source of elevation data)
45 ‘the PRISM’
The sentence was corrected as suggested (L49 The ALOS PRISM DEM is…)
57-60 It appears from what follows, and it is implied here, that individual iDEMs were not used, presumably because they were not available. As they might have been useful for studies of 2011-2014 glacier change (some seasonal function could have been fitted in combination with a linear trend), it should be stated explicitly that iDEMs were not used (and not available?).
It was specified in text, that iDEMs were not available and only TanDEM-X DEM was used in the study (L67-68 The iDEMs, however, are not available to the scientific community, thus only the final product TanDEM-X DEM was used in this study)
60 2011-2014 is not a ‘long period’! I suggest ‘even over a 3-year period, and seasonally’
The sentence was corrected as suggested (L66-67 the inherent variability of glacier surface elevation, seasonally and over a 3-year sampling period.).
72 130 km surprised me, as the dot in Fig. 1A seems somewhat misplaced: according to the scale there, it is plotted just 82 km south of Santiago.
The location of the “Study Area” marker was corrected to better correspond with the map scale. The distance of 130 km was measured in GIS software, so the map had to be adjusted to improve consistency between the text and the figure.
76 ‘supply: glacial’
77 ‘there’ rather than ‘for them’
The sentence was corrected as suggested (L82-83 The heavily populated area of central Chile depends on rivers for water supply: glacial meltwater can be an important source of water there…).
89 and 252; 90, 132 and 264: use of ‘natively’ and ‘native’ here surprised me. Usually ‘native’ is applied to persons, or ‘not introduced’ or ‘occurring in a pure state in nature’ (New Webster’s Dictionary). Would ‘initial’ or ‘original’ or ‘as supplied’ be more appropriate?
The word “native” in relation to the DEM resolutions was replaced with “original” (Table 1, L142 ASTER and SRTM have an original resolution…, L281 accuracies remaining similar to the original resolution, L644 working with datasets with originally lower resolutions.).
138 ‘the glacierized’
The sentence was corrected as suggested and the word “glacierized” was used throughout the text to indicate glacier covered areas (e.g. L37 Aerial Laser Scanning was first used to map glacierized…, Fig. 1 caption D - extent of all glacierized areas, L152-153 Figure 1D presents the whole glacierized…, L640 The uneven performance of TanDEM12 and TanDEM30 over both rocky and glacierized…).
145-146 and Fig.1 Please explain why the glacier outline in C omits some glaciers outlined in D, e..g. in SW and NE: are these glaciers used or of any relevance?
The glacier outline in fig. 1D includes all glaciers and parts of glaciers within the area scanned by ALS. This includes the entire Universidad glacier and only small parts of neighbouring glaciers in NE and NW. In the SE of the area the small glacier is also incompletely covered by the ALS DEM. That’s why they were not subject to mass balance estimations. These areas, although not used for glacier change detection studies, are nevertheless glaciers and thus had to be removed for processing steps which require only non-glaciarised areas. The outline on the Figure 1C encompasses only the studied glacier. The purpose of fig. 1D is to show the Stable Area (SA) extent. This distinction has been clarified in the Figure 1 caption (D - extent of all glacierized areas within the study area, with hillshaded TanDEM DEM 12 m used as basemap. Areas falling outside the green outline and within the ALS DEM extent were considered Stable Area (SA).) and in text (L154-159 The extent of GA is shown on Fig. 1C, while the SA is the area outside of the green-contoured field on Fig. 1D. Although it is seen on Fig. 1D. that Universidad Glacier is not the only glacierized area within the study area, we have decided to omit the fragments of glaciers in the NE, NW and SW of the study area during mass balance assessment, because these limited fragments are not suitable for geodetic mass balance assessment of the whole glaciers.).
Fig. 1B caption: maybe ‘temporal distribution of SAR’
The caption was corrected as suggested (B - temporal distribution of SAR acquisitions used to build TanDEM)
Fig. 1B the ‘clouded areas’ dots in A are black – not white as in key.
The figure was reedited to ensure white colour on both the map and in the key. The colour was proofed in print before submission.
171 Did you ask TanDEM-X admin about the weights? Did they refuse? It seems a strange detail to be secretive about.
Yes, we did reach out to DLR and we were informed that the intermediate DEMs used for averaging to arrive at the final TanDEM-X DEM are not the official delivery product of the DLR and so we cannot be granted access to them.
198 I thought GC was a correction ‘based on slope-aspect’. The distinction between two slope correction methods was not clear in section 3.2: please clarify the distinction between GC and the slope-aspect correction.
Indeed, both correction steps relied on terrain slope and aspect. A new abbreviation SAHC – Slope-Aspect Heatmap Correction – was introduced and used throughout the text to distinguish it from GC (L197-198 Therefore we propose a correction method based on slope-aspect heatmaps (SAHC - Slope-Aspect Heatmap Correction) for correction of these problems.).
235 delete ‘the’
The sentence was corrected as suggested (L252 We use SA-based sc in Table 2 to enable comparisons…)
246 ‘on our quality …’
The sentence was corrected as suggested (L263 The second threshold of 40o is dependent on our quality assessment)
Fig.2 The scale key for E gives ΔZ near the mid-points of colour classes, but F gives them near class boundaries: could both be labelled at the six class boundaries? Also in Fig.3. And could E, F and G have the SAME classes, to permit visual comparison of these graphs? Finally, the five (or six)colours need to be more distinct: the bottom three are hardly distinguishable.
The class boundaries of panels E-G were unified with extreme values set at minimum and maximum of all three datasets. The number of classes was brought to 6 in all cases and label placement was fixed.
Figs. 2 & 3 The initial sentence in each misleadingly implies that all 8 figures are based on SA. Therefore I suggest a change to ‘on SA (A-E), and applied to GA (F-H)’
The captions were corrected as suggested. Results of data analysis and corrections of TanDEM12 relative to the ALS on SA (A-E) and applied to GA(F-H).
277 replace ‘Indeed’ with ‘Also’ - I do not see the connection between the first sentence and the second.
The sentence was corrected as suggested (L294 Also, the spline-surface correction results…)
279-280 ‘lower’ refers to aspect – as this has not previously been discussed, some mention as to why aspect is relevant would be welcome. (It is clear from Fig. 3H that aspects around north are uncommon on the glacier, which faces mainly SSE).
The aspect is important due to the peculiarities of radar scanning. It was mentioned in the methods section and now was elaborated in discussion (L432-442 The accuracy of the Z values deteriorates significantly for slope higher than 40 degrees. The same tendency, also with a threshold of 40o, was noted by Dehecq et al. [46], where DEMs were built by the authors from TanDEM-X SAR acquisitions. There are erroneous areas in TanDEM along the western margin of the glacier (background of Fig. 1B) and their effect on DZ is depicted on Figures 3D-E and 4D-E. Such strong artifacts limit the use of this dataset over steep and rugged terrain. The reason for the artifact’s spatial distribution may be related to the geometry of SAR scanning by TanDEM-X satellites. Eineder [42] proposed a method of assessing the set of slopes reliably visible to a spaceborne radar scanner. He draws a slope/aspect polar plot where the range of slopes useful for InSAR analyses can be depicted, knowing the incidence angle of the radar beam and the angle of satellite orbit. Such analysis does not account for shadowing of areas by the terrain [42], so some areas can lie in radar shadow, despite being in the useful slope range. )
293 due to shifts in the class boundaries between colours, it is not clear that Figs. 2 & 3 support this statement.
This statement is based on Tables 2 and 3 and this was clarified in text (L311 The ALS-TanDEM DZ over the GA is greater than over the SA for both resolutions (Tables 2, 3).)
298-300 reference to Figs. 5 & 6 is desirable here.
References were added (L333-334 The calculated changes obtained from SRTM(Fig. 6) are smaller than those from ASTER (Fig. 7)).
301 ‘DEMs’
The sentence was corrected as suggested (L335 The resolution of the older DEMs…).
316 ‘The case’
The sentence was corrected as suggested (L327-328 …particularly in the case of…).
Section 4.2 I do not accept that comparing ALS with TanDEM-X (2013-2013) gives any information on glacier mass balance (as the heading would imply). ALS-TanDEM=X differences on the glacier depend mainly on the (unknown) weights applied to different TanDEM-X iDEM dates. Perhaps most of lines 293-296, 304-319 and Fig.4 should be given a separate section. All reference to SA is relevant to general quality assessment, not directly to glacier balance, so fuller separation from discussion of the latter would improve clarity.
The fragments in question were merged with the section 4.1. (L311-317 The ALS-TanDEM DZ over the GA is greater than over the SA for both resolutions (Tables 2, 3). This might be expected, as the glacier is a dynamic system, contrary to the surrounding mountain terrain. Thus, multi-temporal averaging of iDEMs is expected to result in an average divergence from the April 2013 surface on the more dynamic part of the scene compared to the stable zone. A greater difference in GA mean DZ is visible when the results of TanDEM30bili and the original TanDEM30 are compared (Table 3). Although the difference of median DZ over SA between the two datasets is only 0.1 m (Table 2), there is almost 1 m of difference between mean DZ of these DEMs on the glacier.). The section was renamed accordingly to “Data correction and quality assessment” (L266).
Table 3 Insert ‘for GA’ ?
The caption was corrected as suggested (Results of DEM subtraction and ice loss calculations for GA.).
Table 3 σp appears here, but neither σc nor σu do. Figs. 4-6 give σc and σ , but is σu nowhere.
The equations 1-4 summarise the Rolstad et al method of calculating uncertainty. The error computed with Rolstad et al method σp, is used in Table 3 and σc in Table 2 to present different uncertainty measured in different contexts. It was clarified in text (L252-254 We use SA-based sc in Table 2 to enable comparisons of precision to other published results. In the Table 3, sp is used as measure of uncertainty for the mean glacier elevation change and all measures derived from it, to account for spatial correlation.).
Fig. 5B You might explain why no TanDEM for SA is included here, whereas it was previously – in Fig.4.
As the SA analysis was relevant only for determination of accuracy of DEMs relative to the reference ALS we have decided to omit TanDEM-ASTER and TanDEM-SRTM SA maps and graphs, as they do not bring any insight into the three DEMs qualities.
378 While a wet surface might be expected for much of the summer, I would expect dry snow after fresh, cold snowfalls at high altitude in parts of spring and autumn.
The number of acquisitions in the early spring and late autumn months is as low as the number of acquisitions in winter months and lower, than in late spring, summer and early autumn. Therefore their influence on the final DEM was assumed to be similar to that of the winter acquisitions. The comment was addressed in text (L409-417 It could be argued that immediately after a summer snowfall the snow would be dry and permeable to the radar signal, influencing even the summer acquisitions. We assume that the snow quickly transitions to a wet state in the conditions of Andean summer. In addition, the summer precipitation in this area is low, so this situation is not frequent enough to influence the TanDEM-X DEM. The early spring and late autumn snowfalls could potentially contribute error in the same way as winter snowfall would do, when temperatures are lower than in summer. The number of acquisitions during these months (April, May and September) is however low, similar to the numbers from winter months (Fig. 1B). Thus, we assume that the dry snow in spring/autumn has a comparable, low, influence on the final TanDEM-X DEMas the winter acquisitions described above.).
391 ‘dataset’
The sentence was corrected as suggested (L436 Such strong artifacts limit the use of this dataset…).
437 EITHER ‘accuracies are poorer’ OR ‘inaccuracies are higher’.
The sentence was corrected to “inaccuracies are higher…” (L510-511 Our calculated inaccuracies are higher).
445 ‘by Puriton …’
The sentence was corrected as suggested (L519 …GPS measurements by Purinton and Bookhagen [26]…).
446 ‘as the measure’
The sentence was corrected as suggested (L521 …GPS was used as the measure of accuracy….).
449 NOT ‘accuracy … below’ but ‘accuracy better than 2 m’
The sentence was corrected as suggested (L524 was determined to be better than 2 m.).
456 ‘Figure 6 ‘ as ASTER is not in Fig.4
The reference was corrected as suggested (L530 Visual analysis of the maps on Figures 6 and 7…).
473 I would like these discontinuities to be illustrated.
A figure was added with Universidad glacier imaged on ASTER RGB composition, ASTER DEM and SRTM DEM to show the presence of ASTER artefacts (Fig. 2). It also served as a proof of lack of cloud cover potentially interfering with ASTER DEM creation (L127-129 Figure 2A shows the study area on a true-color image from ASTER scenes which were used to create the DEM used in the study. No apparent cloud cover is seen over the entire glacier.).
484-485 How does Fig.5 show this? It depends how the dark red stripes near the terminus line up with the medial moraine stripes in Fig. 1D. Cross-sections of this part of each Figure would help support (or refute, therefore drop) this statement.
An analysis of elevation and elevation change between ALS and SRTM DEMs along a cross-section of Universidad tongue was performed. The result is shown on Figure 9 and discussed in the first paragraph of the Section 5.4. (L560-571 The large magnitude of thinning in the lowest part of the glacier points to a retreat trend, in line with the negative mean DZ. According to Barcaza et al. [49], Universidad Glacier lost 0.17 km2 of surface area between 2000 and 2015, hence experiencing a frontal retreat. The thinning rate is not uniform over the lower tongue, where alternating bands of higher and lower DZ are present (Fig. 6). These regions can be related to debris-covered sections of the tongue and elevated bands seen on hillshaded DEMs (Fig. 1C, Fig. 2A). A thin debris cover is known to enhance ablation while a thicker cover insulates the ice and slows down ablation [50]. The debris cover on the tongue and lateral moraines seems to be thick enough to hinder the correct detection of the glacier extent (as mentioned in Section 3.2). The variable pattern of thinning rate along a transect perpendicular to the ice flow in the tongue section (Fig. 9) fits the sections of alternating clean ice and debris in the area. This reveals that the debris cover is sufficiently thick to insulate the underlying ice and decrease surface melt. The result is the formation of clean-ice troughs between debris-covered ridges.).
This allowed to pose a claim of thick debris cover influencing ablation patterns in the area. (L570-571 the debris cover is sufficiently thick to insulate the underlying ice and decrease surface melt. The result is the formation of clean-ice troughs between debris-covered ridges. and L653-654 Debris cover in this area is sufficiently thick to impact ablation patterns and promote the formation of debris-covered ridges.), an interpretation different, that in the previous version of the manuscript.
512 ‘A similar pattern …’
The sentence was corrected as suggested (L596 A similar pattern can be observed…).
523 ‘was a loss of …’
The sentence was corrected as suggested (L602-603 mass balance of Universidad Glacier was a loss of -0.32_0.40 m w.e. [27]).
524 ‘loss’ rather than rate.
The sentence was corrected as suggested (L607-608 long-term loss of 0.44_0.08 m w.e.a‑1.).
527 delete ‘either’
The sentence was corrected as suggested (L611 whether the general 2000-2013 thinning trend is slowing).
547 ‘over the non-glacierized …’ [note; much of the rocky area has been glaciated]
The sentence was corrected as suggested (L631 The median bias of the TanDEM-X DEM over the non-glacierized area was…).
619 delete ‘<’ ?
The symbol was corrected as suggested (L706 sc - standard deviation).
620 ‘correlated’
The word was corrected as suggested (L707 sp - uncertainty of the mean of partially spatially correlated dataset).
621 ‘uncorrelated’
The word was corrected as suggested (L708 su - uncertainty of the mean of spatially uncorrelated dataset).
- Ian S. Evans 26 November 2018

Reviewer 2 Report
In this paper, Podgorski et al. evaluate the performance of TanDEM-X Dem for elevation change detection of mountain glacier demonstrating with an example of Universidad Glacier, located in central Chile. The study uses additional three available DEMs: ASTER, SRTM, and ALS to evalute the TanDEM-X performance. The paper is interesting and the analysis and the treatment of the TanDEM-X DEM data is of good quality. Such study is important for application of the data in glaciological studies and reducing the uncertainty in remote sensing-based measurements. The manuscript can be accepted with minor corrections.
Some recent use of TanDEM data, like Kim and Kim (2017) and Mason et al. (2016) can also be considered in the manuscript.
Even though the manuscript is well presented, the entire manuscript needs minor English editing. Introduction section can be improved logically articulating the statements and the paragraphs. Figures and Tables are fine and appropriate.
Minor suggestions:
L2: Mentions resolutions of ASTER, SRTM, and ALS DEMS in parenthesis
L4-5: Mention 2000-2013?
L5-6: Here can you specify the method instead of writing “an established method….”
L16: “ice elevation change” to “elevation change”
L29-30: Please mention the different methods based on literature, instead of presenting what this paper is going to use. Revise the statement.
L61-62: Revise the objective statement and ambiguity of the statement. Only “glacier change detection” would be fine and thus, not necessary “geodetic mass balance assessment”.
L64-66: The statement only presents some examples. There are also some other studies: e.g., Kim and Kim (2017), Rankl and Braun (2016), Dehecq et al. (2016).
L66-67: Is this statement true. There are papers like, Rankl and Braun (2016) and Dehecq et al. (2016).
L84: Shift section 2.2 to beginning of Method section.
L87-88: In Table 1 and elsewhere in the manuscript: Spatial resolution of the satellite data are based on the pixel size. So, it is not necessary to present the resolution value per pixel (e.g., 12 m/pixel). Only 12 m, 30 m….would be fine.
What do you mean by “Average of 2011-2014” in Time column?
L88: “The UTM Zone 19S projected coordinate system was used with the WGS84 ellipsoid as…..”
L112: …… ASTER DEM V003 (ASTER), a product of METI and NASA ?
L228: statistics term use as italics “r”
L628: change “Bibliography” to “References”
References:
1. Mason, D. C., Trigg, M., Garcia-Pintado, J., Cloke, H. L., Neal, J. C., & Bates, P. D. (2016). Improving the TanDEM-X Digital Elevation Model for flood modelling using flood extents from Synthetic Aperture Radar images. Remote sensing of environment, 173, 15-28.
2. Kim, S. H., & Kim, D. J. (2017). Combined usage of TanDEM-X and CryoSat-2 for generating a high resolution digital elevation model of fast moving ice stream and its application in grounding line estimation. Remote Sensing, 9(2), 176.
3. Rankl, M., & Braun, M. (2016). Glacier elevation and mass changes over the central Karakoram region estimated from TanDEM-X and SRTM/X-SAR digital elevation models. Annals of Glaciology, 57(71), 273-281.
4. Dehecq, A., Millan, R., Berthier, E., Gourmelen, N., Trouvé, E., & Vionnet, V. (2016). Elevation changes inferred from TanDEM-X data over the Mont-Blanc area: Impact of the X-band interferometric bias. IEEE Journal of Selected Topics in Applied Earth Observations and Remote Sensing, 9(8), 3870-3882.
Author Response
REVIEW 2
Open Review
(x) I would not like to sign my review report
( ) I would like to sign my review report
English language and style
( ) Extensive editing of English language and style required
(x) Moderate English changes required
( ) English language and style are fine/minor spell check required
( ) I don't feel qualified to judge about the English language and style
Yes | Can be improved | Must be improved | Not applicable | |
Does the introduction provide sufficient background and include all relevant references? | ( ) | (x) | ( ) | ( ) |
Is the research design appropriate? | ( ) | (x) | ( ) | ( ) |
Are the methods adequately described? | (x) | ( ) | ( ) | ( ) |
Are the results clearly presented? | (x) | ( ) | ( ) | ( ) |
Are the conclusions supported by the results? | ( ) | (x) | ( ) | ( ) |
Comments and Suggestions for Authors
In this paper, Podgorski et al. evaluate the performance of TanDEM-X Dem for elevation change detection of mountain glacier demonstrating with an example of Universidad Glacier, located in central Chile. The study uses additional three available DEMs: ASTER, SRTM, and ALS to evalute the TanDEM-X performance. The paper is interesting and the analysis and the treatment of the TanDEM-X DEM data is of good quality. Such study is important for application of the data in glaciological studies and reducing the uncertainty in remote sensing-based measurements. The manuscript can be accepted with minor corrections.
Some recent use of TanDEM data, like Kim and Kim (2017) and Mason et al. (2016) can also be considered in the manuscript.
Even though the manuscript is well presented, the entire manuscript needs minor English editing. Introduction section can be improved logically articulating the statements and the paragraphs. Figures and Tables are fine and appropriate.
Minor suggestions:
L2: Mentions resolutions of ASTER, SRTM, and ALS DEMS in parenthesis
The resolutions have been added to the abstract (L6-7 Aerial Laser Scanning (ALS)-based dataset from April 2013 (1 m), used as the ground-truth reference, and ASTER V003 DEM and SRTM v3 DEM (both 30 m)).
L4-5: Mention 2000-2013?
The time interval of the analysis has been specified in the abstract (L17-18 Universidad Glacier has been losing mass at a rate of -0.44+/-0.08 m of water equivalent per year between 2000 and 2013.).
L5-6: Here can you specify the method instead of writing “an established method….”
The names of the authors of the method and the year of its publication (Nuth and Kaab (2011)) have been added (L8-9 We use a method of sub-pixel coregistration of DEMs by Nuth and Kääb (2011)).
In addition to the abovementioned corrections, the abstract has been reworked to better reflect the contents of the paper.
L16: “ice elevation change” to “elevation change”
The keyword has been corrected as suggested (L21).
L29-30: Please mention the different methods based on literature, instead of presenting what this paper is going to use. Revise the statement.
The mention of which particular DEMs will be used was removed from this part to make it sound like an introduction to the later summary of ALS and global datasets (L34-35 Laser scanning, radar imaging and photogrammetric analysis of optical stereoimagery are common ways of creating DEMs.).
L61-62: Revise the objective statement and ambiguity of the statement. Only “glacier change detection” would be fine and thus, not necessary “geodetic mass balance assessment”.
The redundant part of the sentence was removed (L69-70 The objective of this work is to determine the suitability of the TanDEM-X DEM for glacier change detection.).
L64-66: The statement only presents some examples. There are also some other studies: e.g., Kim and Kim (2017), Rankl and Braun (2016), Dehecq et al. (2016).
L66-67: Is this statement true. There are papers like, Rankl and Braun (2016) and Dehecq et al. (2016).
The paper by Rankl and Braun (2016) has been cited in our manuscript (no 20) as well as Dehecq et al (2016) (no 46). Authors of both papers use TanDEM-X SAR data to generate DEMs on their own, while the manuscript focuses on TanDEM-X DEM, a ready-made DEM product issued by DLR, which, too, is based on the SAR acquisitions. This part of the introduction presents the earlier works, which analysed TanDEM-X DEM product performance therefore works, where DEMs were made by researchers themselves were not included here.
L84: Shift section 2.2 to beginning of Method section.
The section describing data we have used has been shifted to the Methods section. The sections have been renamed to acknowledge the change: Section 2. is now called “Study area” (L77), while Section 3: “Data and methods” (L90).
L87-88: In Table 1 and elsewhere in the manuscript: Spatial resolution of the satellite data are based on the pixel size. So, it is not necessary to present the resolution value per pixel (e.g., 12 m/pixel). Only 12 m, 30 m….would be fine.
The unit symbol was changed in the whole manuscript.
What do you mean by “Average of 2011-2014” in Time column?
This means that TanDEM-X DEM is a product of averaging of intermediate DEMs acquired between 2011 and 2014. This property of the dataset is discussed in text (L62-65 It is, however, produced by averaging of several DEMs acquired over the years 2011-2014 [23]. The intermediate DEMs (iDEMs) averaged for the final product represent different states of the glacier dependent on season and varying according to a multi-year trend.) and presented in table in a shorter form.
L88: “The UTM Zone 19S projected coordinate system was used with the WGS84 ellipsoid as…..”
The sentence was corrected as suggested (L95-96 The UTM Zone 19S projected coordinate system was used with the WGS84 ellipsoid as the vertical datum.).
L112: …… ASTER DEM V003 (ASTER), a product of METI and NASA ?
The missing “a product of METI and NASA” disclaimer was added in the Acknowledgements section. (L670 ASTER DEM V003 (ASTER) is a product of METI and NASA.).
L228: statistics term use as italics “r”
The symbol was corrected as suggested (L245 a small subset of points (r<300 m)).
L628: change “Bibliography” to “References”
The name of the section has been changed as suggested (L715).
References:
1. Mason, D. C., Trigg, M., Garcia-Pintado, J., Cloke, H. L., Neal, J. C., & Bates, P. D. (2016). Improving the TanDEM-X Digital Elevation Model for flood modelling using flood extents from Synthetic Aperture Radar images. Remote sensing of environment, 173, 15-28.
2. Kim, S. H., & Kim, D. J. (2017). Combined usage of TanDEM-X and CryoSat-2 for generating a high resolution digital elevation model of fast moving ice stream and its application in grounding line estimation. Remote Sensing, 9(2), 176.
3. Rankl, M., & Braun, M. (2016). Glacier elevation and mass changes over the central Karakoram region estimated from TanDEM-X and SRTM/X-SAR digital elevation models. Annals of Glaciology, 57(71), 273-281.
4. Dehecq, A., Millan, R., Berthier, E., Gourmelen, N., Trouvé, E., & Vionnet, V. (2016). Elevation changes inferred from TanDEM-X data over the Mont-Blanc area: Impact of the X-band interferometric bias. IEEE Journal of Selected Topics in Applied Earth Observations and Remote Sensing, 9(8), 3870-3882.
